# Plexin-B2 controls the timing of differentiation and the motility of cerebellar granule neurons

Eljo Van Battum[1†], Celine Heitz-Marchaland[1], Yvrick Zagar[1], Stéphane Fouquet[1], Rohini Kuner[2], Alain Chédotal[1]*

[1]Sorbonne Université, INSERM, CNRS, Institut de la Vision, Paris, France; [2]Pharmacology Institute, Heidelberg University, Heidelberg, Germany

**Abstract** Plexin-B2 deletion leads to aberrant lamination of cerebellar granule neurons (CGNs) and Purkinje cells. Although in the cerebellum Plexin-B2 is only expressed by proliferating CGN precursors in the outer external granule layer (oEGL), its function in CGN development is still elusive. Here, we used 3D imaging, in vivo electroporation and live-imaging techniques to study CGN development in novel cerebellum-specific *Plxnb2* conditional knockout mice. We show that proliferating CGNs in *Plxnb2* mutants not only escape the oEGL and mix with newborn postmitotic CGNs. Furthermore, motility of mitotic precursors and early postmitotic CGNs is altered. Together, this leads to the formation of ectopic patches of CGNs at the cerebellar surface and an intermingling of normally time-stamped parallel fibers in the molecular layer (ML), and aberrant arborization of Purkinje cell dendrites. There results suggest that Plexin-B2 restricts CGN motility and might have a function in cytokinesis.

*For correspondence:
alain.chedotal@inserm.fr

Present address: [†]University Medical Center Utrecht, Department of Translational Neuroscience, Universiteitsweg, Utrecht, Netherlands

## Introduction

Plexins are single-pass transmembrane receptors for Semaphorins regulating cell-cell interactions in normal and pathological contexts (*Pasterkamp, 2012*; *Tamagnone et al., 1999*; *Worzfeld and Offermanns, 2014*). In the developing central nervous system (CNS), Semaphorin/Plexin signaling has been involved in axon guidance and regeneration, neuronal migration (*Pasterkamp, 2012*; *Sekine et al., 2019*; *Yoshida, 2012*), and synaptogenesis (*Hung et al., 2010*; *Kuzirian et al., 2013*; *Molofsky et al., 2014*; *Orr et al., 2017*; *Pecho-Vrieseling et al., 2009*). There is also evidence linking plexins to a variety of neurological diseases such as autism spectrum disorders, multiple sclerosis, Alzheimer's, pathological pain, and spinal cord injury (*Van Battum et al., 2015*; *Binamé et al., 2019*; *Paldy et al., 2017*; *Zhou et al., 2020*).

B-type plexins form a small subclass of plexins, with three members (Plexin-B1, -B2, and -B3) in mammals (*Pasterkamp, 2012*; *Worzfeld et al., 2004*). B-type plexins are not only expressed by neurons but also astrocytes and oligodendrocytes, with some overlapping expression. Like all plexins, their cytoplasmic domain contains a GTPase activating protein (GAP) in which a Rho-binding domain (RBD) is embedded (*Oinuma et al., 2004*; *Seiradake et al., 2016*; *Tong et al., 2007*). Their C-terminal region also interacts with the PDZ (PSD-95, Dlg-1 and ZO-1) domains of two guanine nucleotide exchange factors (GEF), PDZeRhoGEF and leukemia-associated RhoGEF (LARG) (*Pascoe et al., 2015*; *Perrot et al., 2002*; *Seiradake et al., 2016*; *Swiercz et al., 2002*). Plexin dimerization is induced by Semaphorin binding and activates GAP activity, but dimerization was reported to be weaker for Plexin-B2 which might primarily act as a monomer (*Wang et al., 2012*; *Zhang et al., 2015*). Class four transmembrane semaphorins are the main ligands for B-type plexins (*Pasterkamp, 2012*; *Seiradake et al., 2016*; *Tamagnone et al., 1999*) but Plexin-B1 and Plexin-B2 were also shown to interact with the receptor tyrosine kinases ErbB-2 and MET (*Giordano et al., 2002*;

*Swiercz et al., 2004*). It was also recently demonstrated that Plexin-B2 is a receptor for angiogenin, a secreted ribonuclease involved in angiogenesis and amyotrophic lateral sclerosis (*Subramanian et al., 2008*; *Yu et al., 2017*).

Knockout mice for all B-type plexins have been generated but surprisingly, no major brain anomalies have been detected so far in *Plxnb1* (*Deng et al., 2007*) and *Plxnb3* (*Worzfeld et al., 2009*) knockouts. However, altered photoreceptor outer segment phagocytosis in the retina (*Bulloj et al., 2018*) and abnormal migration of Gonadotropin hormone releasing hormone neurons to the hypothalamus (*Giacobini et al., 2008*) were reported in *Plxnb1* knockouts. In contrast, *Plxnb2* knockout mice display severe CNS defects including exencephaly and increased apoptosis (*Deng et al., 2007*; *Friedel et al., 2007*), nociceptive hypersensitivity (*Paldy et al., 2017*), and fear response (*Simonetti et al., 2021*).

The most striking neurodevelopmental defect reported in *Plxnb2* knockout mice is a severe disorganization of the layering and foliation of the cerebellar cortex (*Deng et al., 2007*; *Friedel et al., 2007*; *Maier et al., 2011*; *Worzfeld et al., 2014*). The cerebellum contains a limited and well-characterized number of neuronal types (about 9) and its cortex has only three layers: the inner granular layer (IGL), the deepest one, contains granule cells (CGNs), the Purkinje cell layer, and most externally, the molecular layer, which hosts Purkinje cell dendrites, CGN axons and two types of interneurons, the stellate and basket cells (*Sotelo, 2011*; *Voogd, 2003*). Purkinje cell axons are the sole output of the cerebellar cortex. Cerebellar neurons originate from the ventricular zone of the cerebellum primordium, except CGNs and unipolar brush cells that arise in the so-called upper rhombic lip (*Leto et al., 2016*). In the mouse brain, cerebellar CGNs account for about half of the neurons all generated after birth from progenitors localized in a transient neuroepithelium, the external granular layer (EGL), occupying the surface of the cerebellum until about the third postnatal week (*Chédotal, 2010*). The EGL develops embryonically as CGN precursors migrate from the rhombic lip to colonize the surface of the cerebellar anlage (*Miale and Sidman, 1961*). Symmetrical division amplifies the pool of precursors until birth, after which they start dividing asymmetrically to generate CGNs. Post-mitotic CGNs segregate from dividing precursors and move inward splitting the EGL into two sublayers: the outer EGL (oEGL) containing proliferating cells and the inner EGL (iEGL) containing newly born CGNs. In the iEGL, CGNs migrate tangentially (parallel to the cerebellar surface), grow two processes (their presumptive axons or parallel fibers) and adopt a bipolar morphology (*Komuro et al., 2001*; *Cajal, 1909*). CGNs next extend a third process perpendicular to the surface, which attaches to Bergmann glia fibers and guide the inward radial migration of CGNs across the molecular layer to the inner granule cell layer. Strikingly, Plexin-B2 is only expressed in the oEGL and downregulated in post-mitotic CGNs (*Friedel et al., 2007*). The phenotypic analysis of two *Plxnb2* complete knockout lines showed that the lack of Plexin-B2 maintains some migrating CGNs in a proliferating state which leads to a massive disorganization of cerebellar cortex layers (*Deng et al., 2007*; *Friedel et al., 2007*). This phenotype has been also observed in a conditional knockout lacking Plexin-B2 in CGN precursors (*Worzfeld et al., 2014*). However, the exact consequences of Plexin-B2 deficiency on CGN development are unknown and were not studied at a cellular level. Here, we used a combination of 3D imaging, in vivo electroporation and live imaging to study the development of CGNs in cerebellum-specific conditional knockouts. We show that the transition from a multipolar to a bipolar morphology, the migration speed and CGN axon distribution are altered in absence of Plexin-B2.

## Results

### Cerebellum-specific inactivation of Plxnb2 affects foliation and lamination

In the mouse cerebellum, which matures during the first three postnatal weeks, proliferating CGN precursors in the EGL, which can be labeled using 5-Ethynyl-2'-deoxyuridine (EdU), express Plexin-B2 (*Friedel et al., 2007*; *Worzfeld et al., 2004*; *Figure 1A,B*). Plexin-B2 expression diminishes when CGNs start to migrate in a tangential direction (*Figure 1B*). As the EGL resorbs (*Figure 1C*) and becomes depleted of CGN precursors and post-mitotic CGNs, Plexin-B2 expression progressively disappears (*Figure 1C*). As previously shown, a small fraction of *Plxnb2⁻/⁻* mutant mice, bred in the

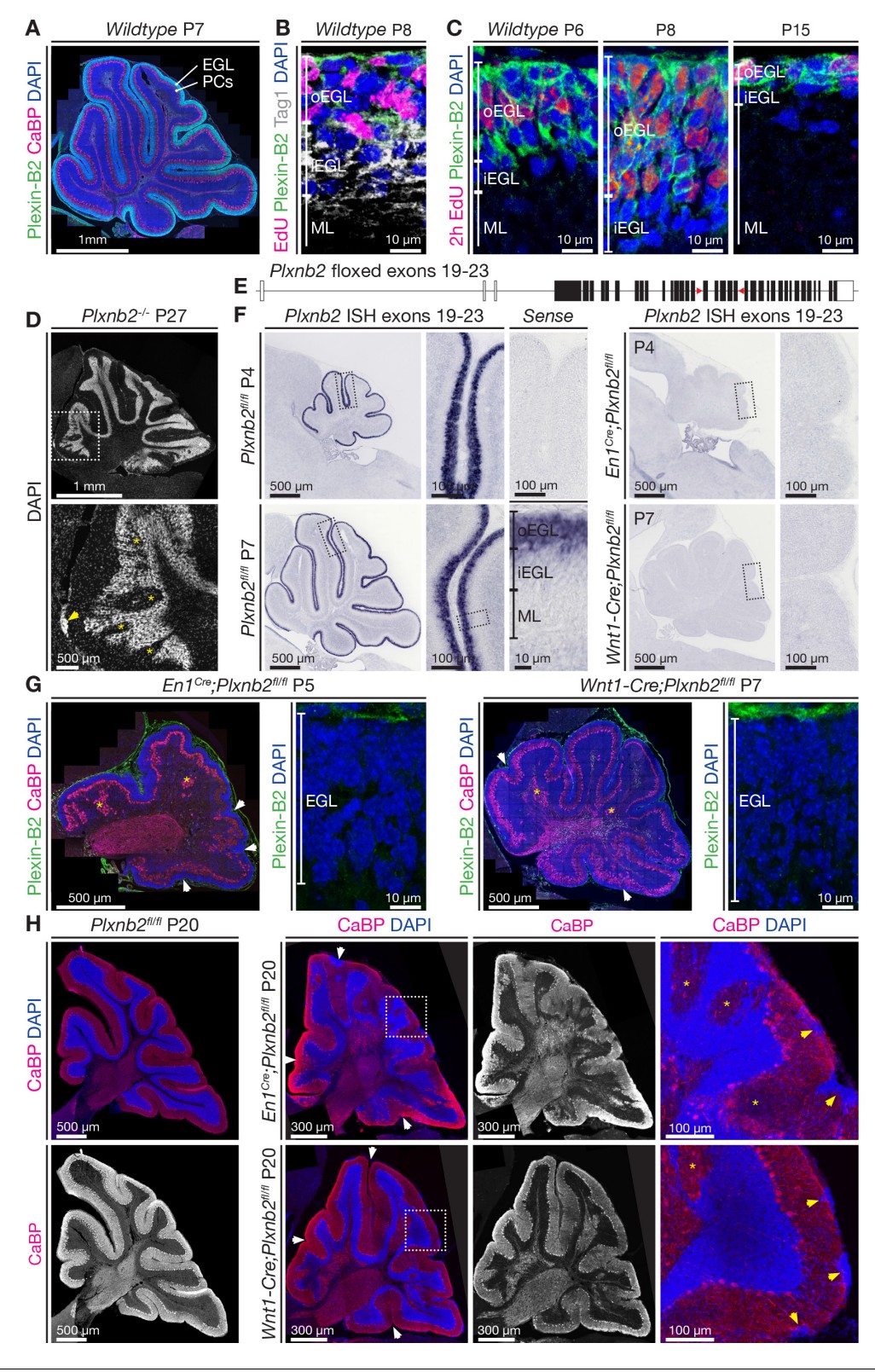

**Figure 1.** Plexin-B2 expression and generation of cerebellum-specific *Plxnb2* conditional knockout models. (A, B, C), Plexin-B2 protein distribution in the cerebellar cortex during different stages of postnatal development. (A) Plexin-B2 immunostaining on cryostat sections immunolabeled with the Purkinje cell (PC) marker Calbindin (CaBP) and counterstained with DAPI shows that Plexin-B2 is expressed in the external granule layer (EGL). (B) Plexin-B2 immunoreactivity coincides with EdU (injected 2 hr prior to fixation to visualize proliferating cells) showing that this receptor is restricted to

*Figure 1 continued on next page*

Figure 1 continued

proliferating cerebellar granule neurons (CGNs) in the outer external granular cell layer (oEGL). It is downregulated in Tag1$^+$ postmitotic CGNs in the inner EGL (iEGL). (C) High-magnification images show Plexin-B2 expression in the oEGL (stained with EdU), which regresses between P6 and P15. (D) Sagittal section of P27 cerebellum *Plxnb2$^{-/-}$* (full knockout) cerebellum stained with DAPI. The structure and layers of the cerebellar cortex are disorganized. Clear gaps in the internal granule layer structure can be observed (yellow asterisks), as well as patches of cells that accumulated at the cerebellar surface (arrowhead). (E) Schematic representation of the genomic *Plxnb2* sequence of the conditional *Plxnb2* mutant described in **Deng et al., 2007**. The *loxP* sites flanking exons 19–23 are depicted with red triangles. *Plxnb2$^{fl/fl}$* conditional mutant mice were crossed with *En1$^{Cre}$* or *Wnt1-Cre* mice. (F) In situ hybridization, on cerebellar sections at P4 and P7, with a probe recognizing the floxed exons of the *Plxnb2* gene. Sections incubated with *sense* probe are devoid of signal. In *cre*-negative *Plxnb2$^{fl/fl}$* control mice, *Plxnb2* mRNA is only detected in the oEGL. In both *En1$^{Cre}$;Plxnb2$^{fl/fl}$* and *Wnt1-Cre;Plxnb2$^{fl/fl}$* littermates, *Plxnb2* mRNA is deleted from the oEGL. (G) Plexin-B2 immunostaining on sagittal cerebellar sections of *En1$^{Cre}$;Plxnb2$^{fl/fl}$* (P5) and *Wnt1-Cre;Plxnb2$^{fl/fl}$* (P7) animals shows the absence of Plexin-B2 protein in the EGL. Sections were also labeled with anti-CaBP antibodies and DAPI. Impaired cerebellar foliation (white arrowheads) and Purkinje cell islands (yellow asterisks) are observed in both conditional knockouts. (H) P20 sagittal cerebellar sections immunostained for CaBP and counterstained with DAPI. Both *En1$^{Cre}$;Plxnb2$^{fl/fl}$* and *Wnt1-Cre;Plxnb2$^{fl/fl}$* conditional knockouts phenocopy the cerebellar defects found in *Plxnb2$^{-/-}$* mutants. White arrowheads mark altered foliation, whereas yellow arrowheads in the magnified panels show surface accumulations of CGNs. Yellow asterisks indicate Purkinje cell islets. *En1$^{Cre}$;Plxnb2$^{fl/fl}$* mice display the *Plxnb2* phenotype to a greater extent. Scale bars: (A) 1 mm. (B, C) 10 µm. (D) Low magnification 1 mm, high magnification 500 µm. (F) Low-magnification overview panels: 500 µm, high-magnification panels: 100 µm. (G) Overview panels: 500 µm, high-magnification EGL panels: 10 µm. (H) Low-magnification panels 300 µm, high-magnification panels 100 µm.

CD1 background, survive and display severe cerebellar disorganization (**Friedel et al., 2007**; **Maier et al., 2011**; **Figure 1D**).

To circumvent lethality of the *Plxnb2* full knockout mouse model, a *Plxnb2* gene with floxed exons 19–23 (**Figure 1E**, **Worzfeld et al., 2014**) was crossed with *Engrailed En1$^{Cre}$* and *Wnt1-Cre* lines (see Materials and methods). Under the *En1* promoter, *Cre* is driven in all mesencephalon and rhombomere one leading to expression in the midbrain, a part of the hindbrain and the entire cerebellum (**Kimmel et al., 2000**; **Zervas et al., 2004**). Under the *Wnt1* promoter, *Cre* is initially expressed in the cerebellum by CGNs and sparsely in other cell types in the cerebellum (**Nichols and Bruce, 2006**). Both *En1$^{Cre}$;Plxnb2$^{fl/fl}$* and *Wnt1-Cre;Plxnb2$^{fl/fl}$* lines were viable. In situ hybridization with a *Plxnb2* probe encompassing exons 19–23, confirmed that, unlike in *Plxnb2$^{fl/fl}$* controls, *Plxnb2* expression was undetectable in the EGL of *En1$^{Cre}$;Plxnb2$^{fl/fl}$* and *Wnt1-Cre;Plxnb2$^{fl/fl}$* mice (**Figure 1F**). In addition, the EGL of *En1$^{Cre}$;Plxnb2$^{fl/fl}$* (P5) and *Wnt1-Cre;Plxnb2$^{fl/fl}$* (P7) cerebellum was completely devoid of Plexin-B2 protein immunoreactivity (**Figure 1G**). Importantly, a severe disorganization of the foliation and layering of the cerebellum was observed in both conditional knockouts (**Figure 1F,G,H**) which phenocopied what has been previously reported in the *Plxnb2* null knockout (**Figure 1B**, **Friedel et al., 2007**). The mutant Purkinje cell layer (visualized using anti-Calbindin (CaBP) immunostaining, **Figure 1H**) showed the characteristic Purkinje cell islets, and the IGL appeared very disorganized. We focused for the rest of the study on *En1$^{Cre}$;Plxnb2$^{fl/fl}$* knockouts as En1 has a more restricted expression than Wnt1 (which is expressed in all sensory ganglia), and *En1$^{Cre}$;Plxnb2$^{fl/fl}$* mice displayed a more severe cerebellar phenotype. Moreover, midbrain defects were reported in *Wnt1-Cre* mice (**Lewis et al., 2013**).

We next studied the postnatal development of cerebellum lamination and folding in *En1$^{Cre}$; Plxnb2$^{fl/fl}$* mutants. A striking delay in the formation of the cerebellar fissures was observed in *En1$^{Cre}$;Plxnb2$^{fl/fl}$* mutant mice, which was already visible at birth (**Figure 2A**). In normal mice, the principle cerebellar fissures start to appear from E17.5 onwards (**Sudarov and Joyner, 2007**). Whereas in control animals the six principal cerebellar fissures were clearly visible at P0, the cerebellum remained smooth in the *Plxnb2* mutant (**Figure 2A**). Even if most fissures eventually emerged after P4 in *En1$^{Cre}$;Plxnb2$^{fl/fl}$*, they were not as deep as in *Plxnb2$^{fl/fl}$* controls (**Figure 2A**). Another hallmark of the phenotype described for *Plxnb2$^{-/-}$* mice are ectopic islets of Purkinje cells in the midst of the IGL (**Friedel et al., 2007**). In *En1$^{Cre}$;Plxnb2$^{fl/fl}$* cKO cerebella displaced PCs were first detected at P2 but became more conspicuous from P4 (**Figure 2A**).

To better comprehend the cerebellar alterations in *En1$^{Cre}$;Plxnb2$^{fl/fl}$* cerebellum, we performed 3D-light sheet fluorescence microscopy (LSFM) of iDISCO+ cleared brains (**Renier et al., 2016**). Nuclear TO-PRO-3 staining confirmed the altered fissure formation in P4 cerebellum and also showed the development of additional folds of the IGL perpendicular to the main fissures (**Figure 2B**). Despite the aberrant folds, CGNs and Purkinje cell lamination was grossly preserved (**Video 1**).

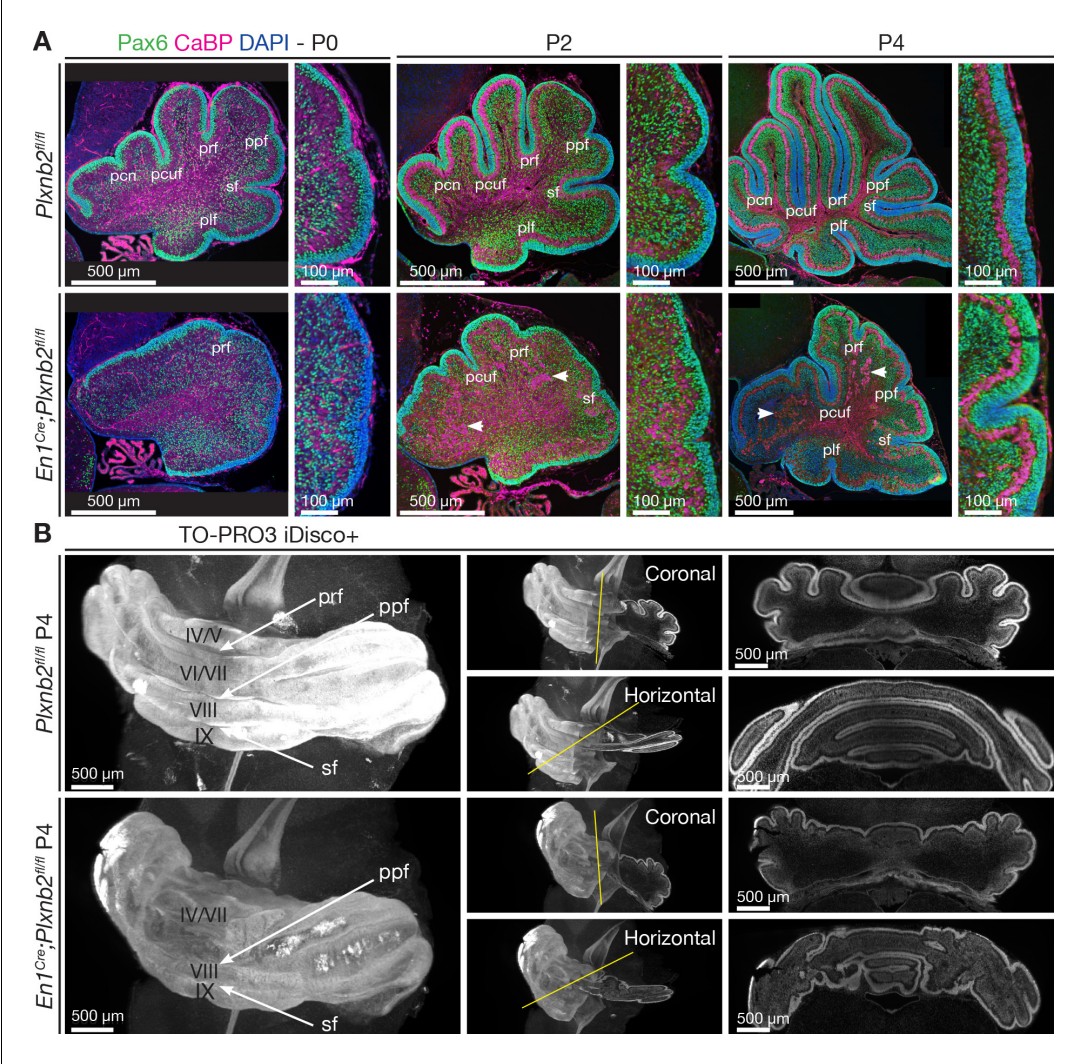

**Figure 2.** Developmental time course of cerebellar *Plxnb2* phenotype. The time course of cerebellar foliation and lamination during early postnatal cerebellar development is delayed in *Plxnb2* conditional knockout. (**A**) Pax6 immunostaining labels both pre- and postmitotic CGNs in the developing cerebellum, and Calbindin (CaBP) labels Purkinje cells. In controls, many cerebellar fissures have formed at P0, and deepen further at P2 and P4, whereas the cerebellum of *En1^Cre;Plxnb2^fl/fl* mutant displays a very shallow primary fissure (prf) at P0 and aberrant fissure development over time. Furthermore, ectopic Purkinje cell islets (arrowheads) are observed in *Plxnb2* mutant internal granule layer. (**B**) 3D-Light sheet microscope imaging of TO-PRO-3 stained and iDISCO+ cleared (see Materials and methods) P4 cerebellum illustrating the foliation delay in *Plxnb2* conditional KO. Right panels are optical sections (coronal or horizontal) through 3D-reconstructed images. *Plxnb2* mutants develop aberrant shallow fissures and additional folia in different orientations. Abbreviations: pfr: primary fissure, ppf: prepyramidal fissure, sf: secondary fissure, pcn: precentral fissure, pcuf: preculminate fissure, pfl: posterolateral fissure. Scale bars: overview panels (**A, B**): 500 µm, magnifications in (**A**): 100 µm.

As cerebellar development progresses, the *Plxnb2* mutant phenotype became more severe. In P14 controls, cerebellar fissures were fully developed, the EGL was almost absent and IGL structure was very smooth and homogeneous (*Figure 3A*). In contrast, *Plxnb2* mutants showed patches of CGNs remaining at the cerebellar surface, an IGL structure with many invaginations in different orientations, and original fissures could not be defined easily (*Figure 3B*, *Video 2*). The aberrant IGL structure and the patches of granule cells at the surface persisted in adulthood, well after cerebellar development was completed (*Figure 3C,D*, *Video 3*). All aberrant IGL folds in the *Plxnb2* mutant were lined with a monolayer of Purkinje cells (*Video 4*). These 3D-data convey that the Purkinje cell 'islets' observed in *Plxnb2* mutant cerebellar sections actually correspond to stretches of Purkinje cells that line the heavily corrugated, but still continuous IGL.

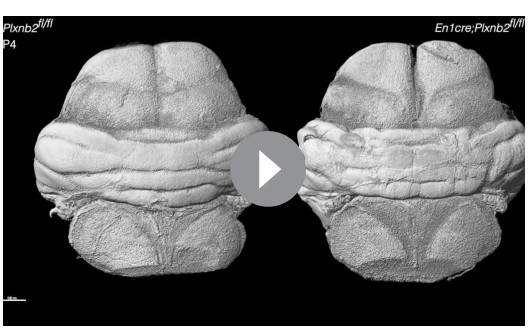

**Video 1.** 3D movie of P4 iDISCO+ cleared *Plxnb2*^fl/fl^ and *En1*^Cre^*;Plxnb2*^fl/fl^ cerebella. All cell nuclei are stained with TO-PRO-3, Pax6 and FoxP2-staining is used to visualize CGNs and Purkinje cell bodies, respectively.

https://elifesciences.org/articles/60554#video1

Because *Plxnb2* mutant cerebella seemed to be smaller than controls on sections, we next analyzed the cerebellar volume in 3D. Indeed, a limited, but significant reduction was observed throughout cerebellar development (*Figure 3—figure supplement 1A*).

*Plxnb2* mutant did not display any noticeable motor or behavioral defects and their performance on an accelerating rotarod was similar to control mice (*Figure 3—figure supplement 1B*).

## Plxnb2 mutant CGNs disorganize the EGL and proliferate slightly different

Since in the cerebellum, Plexin-B2 is only expressed in proliferating CGN precursors in the EGL, we characterized the cellular organization of this layer in more detail. We visualized the outer layer of proliferating CGNs in the EGL by injecting P9 mouse pups with EdU 2 hr before perfusing them (*Figure 4A*). Purkinje cells were immunostained with anti-CaBP (*Figure 4A*). In controls, whereas EdU+ CGN precursors were usually confined to the thin outer EGL (oEGL), they were more dispersed in *Plxnb2* mutants (*Figure 4A*). In addition, the developing ML was much thinner (*Figure 4A*). We next performed double immunostaining for Ki67, a marker of proliferating precursors, and Tag1 (Transient axonal glycoprotein 1, also known as Contactin-2), which labels tangentially migrating CGNs in the iEGL (*Figure 4B*). In control P9 cerebellum, both markers were segregated (*Figure 4B*), whereas in *Plxnb2* mutants, CGN precursors lost their confinement to the oEGL and intermingled with tangentially migrating Tag1+ CGNs in the iEGL (*Figure 4B*). However, as Ki67 and Tag1 are expressed in different cell compartments (nucleus and cell surface respectively), we could not determine if some of the Tag1+ cells were also Ki67+.

These results lead us to investigate potential differences in *Plxnb2* mutant CGN precursor proliferation. Mice were given a short (2 hr before fixation at P8) or a long (24 hr before fixation at P8) EdU pulse, and the number of EdU+ cells in the EGL was counted (*Figure 4C*, *Figure 4—figure supplement 1A*). No difference in the amount of EdU+ cells was detected for either time of EdU administration, and there was no difference in the uptake of EdU by dividing cells over time (*Figure 4C*). In addition, the quantification of the amount of EdU+ CGNs in the EGL (EdU injected 2 hr before brain collection) that colocalized with Ki67 immunostaining did not show a difference in proliferation rate in *Plxnb2*-mutant brains (*Figure 4—figure supplement 1B*). Since it is estimated that CGN precursors divide approximately every 20 hr (*Espinosa and Luo, 2008*), cerebellum sections of pups injected with EdU were stained after 24 hr for Phospho-histone H3 (H3P), an M-phase marker. This enabled us to quantity the proportion of cells that took up EdU the day before (and theoretically should have ended division hours ago) and were still in their cycle. Interestingly, we observed a significant increase in the percentage of cells double-positive for EdU and H3P (*Figure 4D*). This implies that the cell cycle for *Plxnb2* mutant CGNs is slightly longer. Together, these results suggest that there is probably no alteration of cell cycle progression in absence of Plexin-B2 although more experiments will be required to determine if the M phase is affected.

## Migrating Plxnb2 mutant CGNs show different morphology and proliferation markers

The high cell density of the EGL makes it difficult to follow the morphological changes that developing CGNs undergo during the different steps of their development. To follow newborn CGNs throughout their developmental sequence, we targeted CGNs in the EGL of P7 mouse pups with GFP using in vivo electroporation (*Figure 5A*). By adapting a tripolar electrode electroporation technique developed for embryos (*dal Maschio et al., 2012*) to postnatal mice, we could reproducibly target a wide area of the dorsal cerebellum. With this method 99.6% of all targeted GFP+ cells in the cerebellum were CGNs and co-expressed Pax6 (*Figure 5—figure supplement 1A*). The

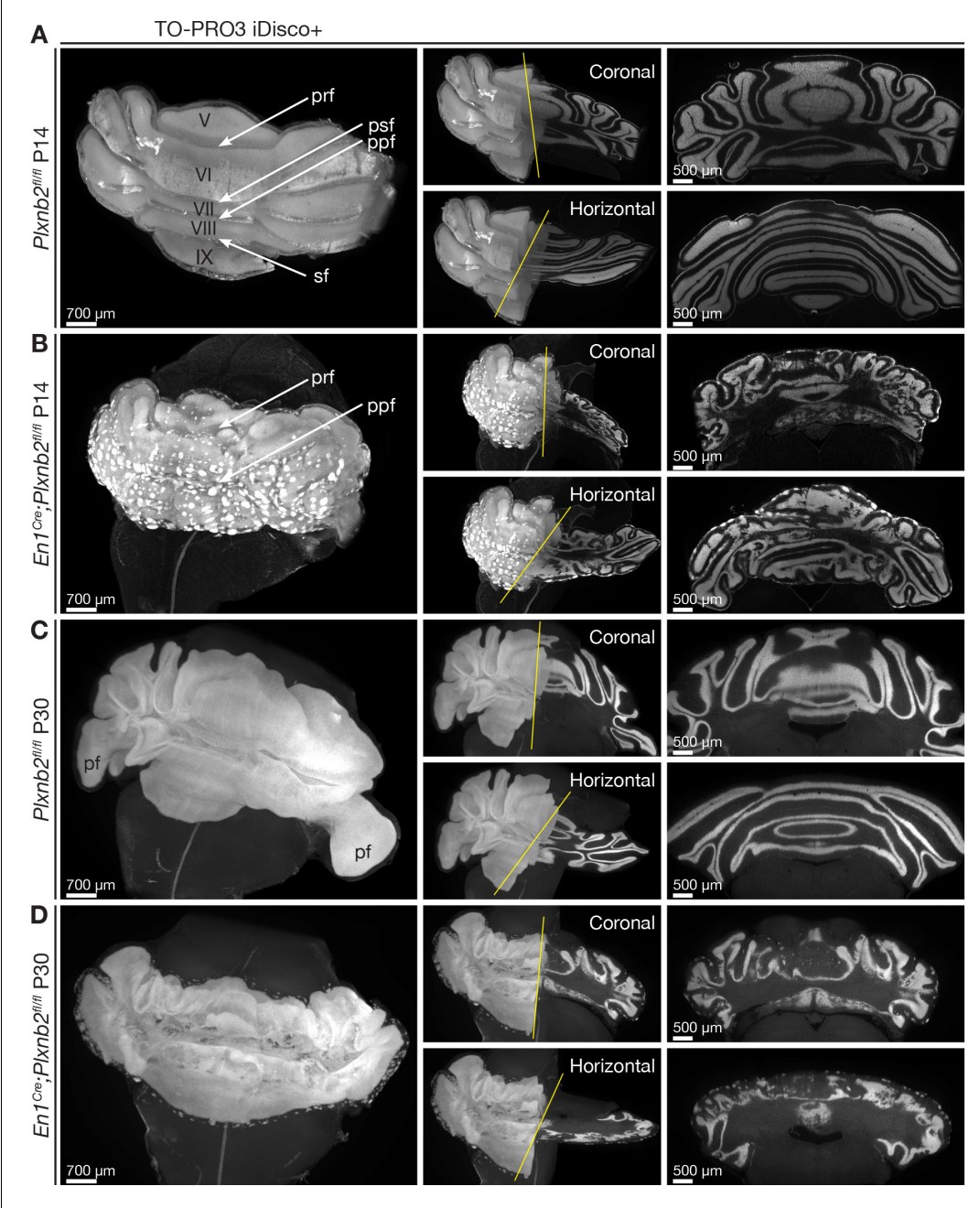

**Figure 3.** Ruffled IGL and ectopic CGN patches in cerebellum-specific *Plxnb2* mutant. (A– D) Whole-mount TO-PRO-3 staining of P14 and P30 cerebella from *Plxnb2^fl/fl^* and *En1^Cre^;Plxnb2^fl/fl^* littermates cleared with iDISCO+. In 3D, TO-PRO-3 staining mainly reveals the structure of the cell-dense IGL. *Plxnb2^fl/fl^* control cerebella (A) display a very smooth IGL. A thin layer of EGL remains at P14 but not at P30 (C). In P14 *En1^Cre^;Plxnb2^fl/fl^* mice, the regressing EGL contains ectopic clusters of CGNs (B) that remain at P30 (D). In addition, *Plxnb2* mutant IGL (B, D) shows many invaginations in different directions, independent of normal fissure orientation. Although some fissures are clearly formed and visible (prf, ppf), many others are absent. The paraflocculus (pf), present in P30 control, was lost during dissection in the *Plxnb2* mutant. Greek numbers indicate cerebellar lobes. Scale bars: 700 µm for the 3D images, 500 µm for the coronal and horizontal sections. Pfr: primary fissure, psf: posterior superior fissure, ppf: prepyramidal fissure, sf: secondary fissure. *Figure 3—figure supplement 1A* shows quantification of 3D cerebellar volume (*Figure 3—figure supplement 1—source data 1*). The online version of this article includes the following source data and figure supplement(s) for figure 3:

**Figure supplement 1.** Difference in cerebellar volume but not motor function.

**Figure supplement 1—source data 1.** Cerebellar volume and function.

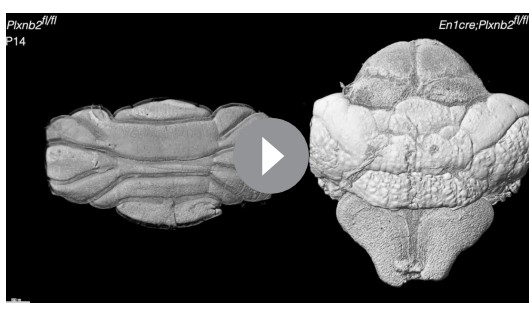

**Video 2.** 3D movie of P14 iDISCO+ cleared *Plxnb2^{fl/fl}* and *En1^{Cre};Plxnb2^{fl/fl}* cerebella. All cell nuclei are stained with TO-PRO-3.

https://elifesciences.org/articles/60554#video2

developmental sequence of CGNs is stereotypically phased, and by collecting cerebella at specific time-points post-electroporation, we could study their morphological evolution, from precursors to tangential migrating cells, radial migrating cells, and maturing CGNs in the IGL (*Figure 5B*). After 24 hr, most electroporated GFP^+ CGNs were in the tangential phase, while some still resided in the oEGL. Two days after electroporation, GFP^+ CGNs had started radial migration, and extended parallel fibers (adopting a characteristic T-shape, *Figure 5B*). Subsequently, 1week after electroporation, all GFP^+ CGNs had reached the IGL and started the process of dendritogenesis. Eventually, all GFP^+ CGNs displayed their stereotypical morphology with a small cell body bearing 3–4 claw-shaped dendrites (*Figure 5B*).

By comparing the initial steps of postmitotic CGN development, we found a significant reduction in the length of the processes (and future parallel fibers) extended by tangentially migrating CGNs in *Plxnb2* mutants compared to controls (*Figure 5C,D*). Cell body size and shape for proliferating and tangentially migrating CGNs was similar in both genotypes (*Figure 5—figure supplement 1B*). As implied by the EdU and Ki67/Tag1 immunohistochemistry (*Figure 4A,B*), we observed that GFP^+ CGN precursors intermingled with migrating bipolar CGNs in the EGL of *Plxnb2* mutant animals (*Figure 5C*). Indeed, quantification of the location of multipolar and bipolar CGNs in the outer, inner EGL or ML, shows that multipolar and bipolar CGNs in the *Plxnb2* mutant were spread throughout the EGL and that bipolar CGNs sometimes even resided in the ML (*Figure 5—figure supplement 1C*). Intriguingly, in *Plxnb2* mutants, a fraction of the bipolar and tangentially migrating GFP^+ CGNs, were also labeled with EdU (which only labels dividing cells), administered 2 hr before fixation. By contrast, only a very small fraction of tangentially migrating bipolar GFP^+ CGNs were found in controls (*Figure 5C,E*). When combining the EdU staining with H3P to mark acutely dividing cells, we could confirm that, although rare, some of the bipolar GFP^+/EdU^+ were indeed proliferating (*Figure 5—figure supplement 1D*). Some bipolar GFP^+ CGNs also co-expressed H3P and Tag1 (*Figure 5F*). This suggests that in *Plxnb2* mutants, the CGN switch from proliferation to tangential migration is altered and that these two phases are not spatio-temporally separated anymore. To further support this hypothesis, we next analyzed GFP^+ CGNs 48 hr after electroporation. Again, there was a significant reduction in the proportion of CGNs that had initiated radial migration in *Plxnb2* mutants (*Figure 5G*). This, together with the slight delay in M-phase (*Figure 4D*), suggest that *Plxnb2*-deficient CGNs might remain longer in both their proliferative and tangential phases.

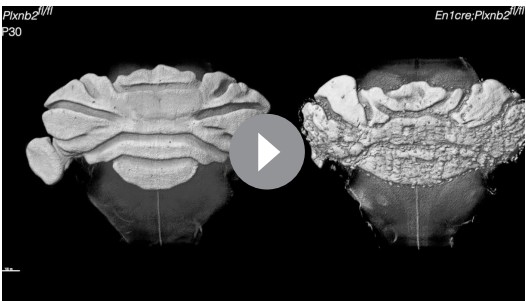

**Video 3.** 3D movie of P30 iDISCO+ cleared *Plxnb2^{fl/fl}* and *En1^{Cre};Plxnb2^{fl/fl}* cerebella. All cell nuclei are stained with TO-PRO-3.

https://elifesciences.org/articles/60554#video3

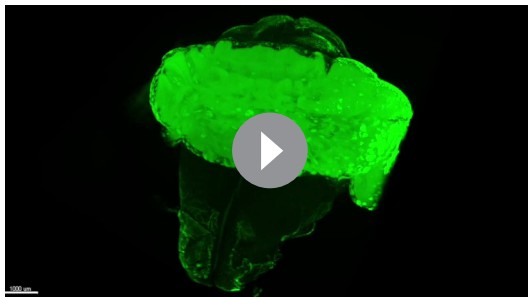

**Video 4.** 3D movie of P20 iDISCO+ cleared *En1^{Cre};Plxnb2^{fl/fl}* cerebellum stained with TO-PRO-3 for all cell nuclei and FoxP2 to visualize Purkinje cell bodies.

https://elifesciences.org/articles/60554#video4

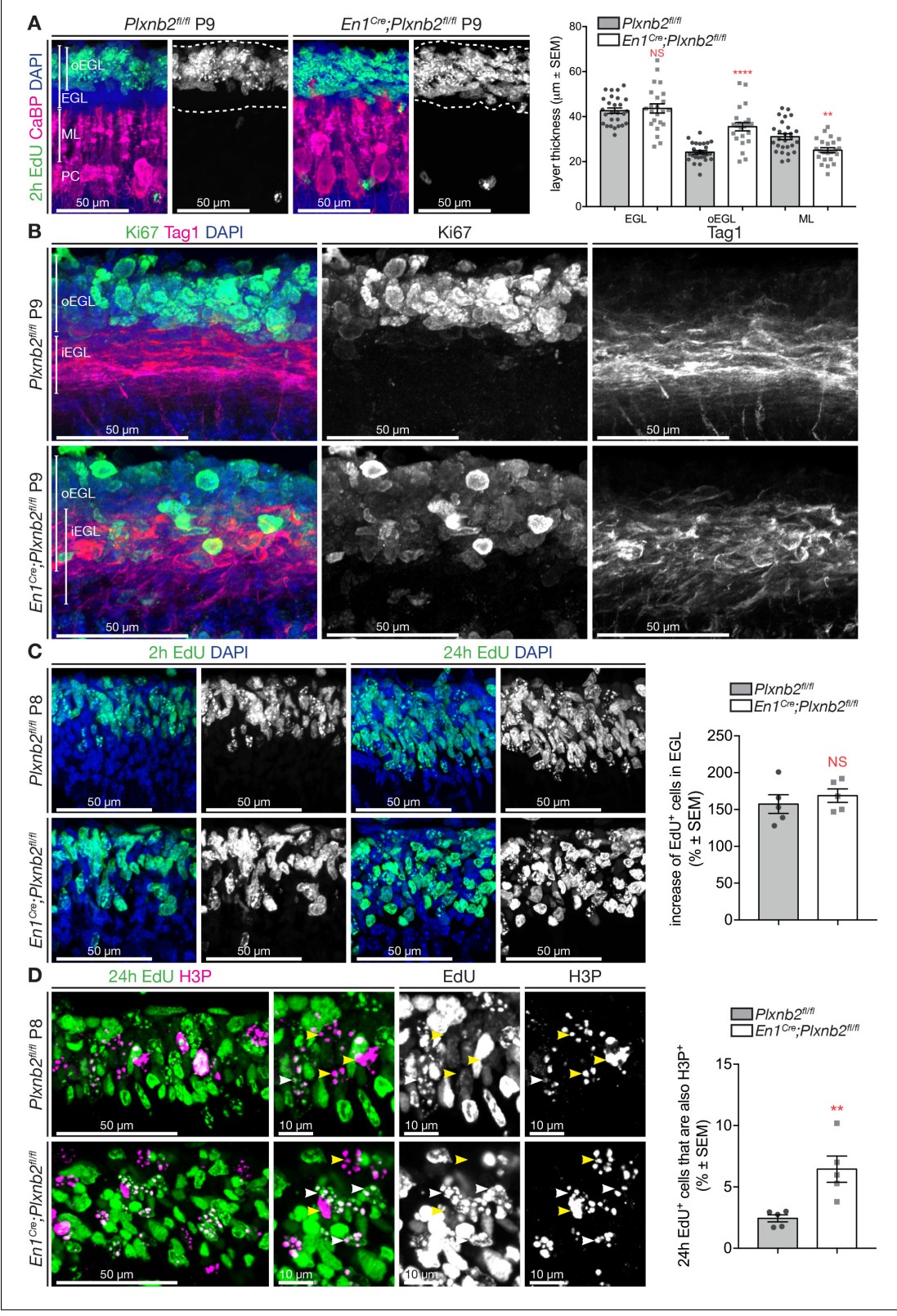

**Figure 4.** Proliferating CGNs intermingle with migrating CGNs and have a longer cell-cycle. (**A**) Coronal sections of P9 cerebella of *Plxnb2^{fl/fl}* control and *En1^{Cre};Plxnb2^{fl/fl}* littermates injected with EdU 2 hr before perfusion. EdU labels proliferating CGN precursors and Calbindin (CaBP) immunostaining labels Purkinje cells. Sections were counterstained with DAPI. In control, proliferating CGNs (EdU⁺) are restricted to the outer layer of the EGL (oEGL). In *Plxnb2* mutant, EdU⁺ CGN precursors are found throughout the EGL. The developing molecular layer, containing CaBP⁺Purkinje cell

*Figure 4 continued on next page*

*Figure 4 continued*

dendrites, is thinner in *Plxnb2* mutant. The graph shows the quantification of the thickness of the EGL, oEGL, and molecular layer (ML). Error bars represent SEM. EGL: 42.63 ± 1.19 µm in ctl *vs*. 43.62 ± 2.04 µm, in mut, MWU(295) p=0.77, NS: not significant. oEGL 24.78 ± 0.42 µm in ctl *vs*. 35.32 ± 0.76 µm in mut. MWU(70) p<0.0001. ML: 32.04 ± 0.71 µm in ctl *vs*. 27.44 ± 0.64 µm in mut. MWU(159) p=0.027. (*Figure 4—source data 1*) (B) Coronal sections of P9 cerebella from *Plxnb2^{fl/fl}* control and *En1^{Cre};Plxnb2^{fl/fl}* littermates, immunostained for Ki67 and Tag1. Ki67 labels proliferating CGN precursors in the oEGL and Tag1 postmitotic CGNs that migrate tangentially in the inner EGL (iEGL). These two populations of precursors and postmitotic neurons are strictly separated in controls, whereas they intermingle in *Plxnb2* mutants. (C) Sagittal sections of P8 cerebella from *Plxnb2^{fl/fl}* and *En1^{Cre};Plxnb2^{fl/fl}* littermates injected with EdU 2 hr, or 24 hr prior to fixation. EdU⁺ cells were counted and averaged from three sections per animal from five ctl and five mut animals. No difference in the production of new CGNs between 2 and 24 hr of EdU were observed. Graph shows the percentage of EdU⁺ cells in the EGL after 24 hr compared to 2 hr (ctl 157.4 ± 12.64% *vs*. mut 168.8 ± 9.22%, MWU(9), p=0.55, not significant). Error bars represent SEM. (*Figure 4—source data 1*) Graph in *Figure 4—figure supplement 1A* shows that there is no difference in the raw amount of EdU⁺ cells per µm³ after 2 hr or 24 hr post-injection as counted from these sections (*Figure 4—figure supplement 1—source data 1*). (D) Immunohistochemistry of sagittal sections of P9 cerebella from *Plxnb2^{fl/fl}* and *En1^{Cre};Plxnb2^{fl/fl}* littermates injected with EdU 24 hr prior to fixation. EdU labels cells that started their division cycle in the last 24 hr while H3P staining labels dividing cells. The graph shows the amount of cells in the EGL that are both EdU and H3P positive is higher in the *Plxnb2* mutant. Error bars represent SEM. Ctl: 2.44 ± 0.29% *vs*. mut: 6.45 ± 1.07%. MWU(0) p=0.0079. (*Figure 4—source data 1*) Scale bars: 50 µm in (A, B, C) and (D), 10 µm in high-magnification panels of (D).

The online version of this article includes the following source data and figure supplement(s) for figure 4:

**Source data 1.** Layer thickness and proliferation markers in EGL.

**Figure supplement 1.** No difference in amounts of EdU⁺ and H3P⁺ CGNs in EGL.

**Figure supplement 1—source data 1.** No difference in amounts of EdU⁺ and H3P⁺ CGNs in EGL.

As soon as *Plxnb2* mutant CGNs started radial migration, their morphology closely resembled that of control CGNs. Differences in leading process length were no longer observed, and their ascending axons in the ML were of comparable length (*Figure 5—figure supplement 1E*). At this radial stage in both control and mutant GFP⁺ CGNs, proliferation markers were seldom observed (*Figure 5G*). Conversely, during radial migration, CGN cell bodies had a slightly more circular shape in *Plxnb2* mutants (*Figure 5—figure supplement 1E*). Moreover, in *Plxnb2* mutants, CGNs in the IGL acquired their stereotypical CGN morphology with a cell body of 7–8 µm in diameter bearing 3–4 claw-shaped primary dendrites, slightly faster than in controls (*Figure 5—figure supplement 1F, G*). Together, this suggests that in the IGL, CGNs might differentiate faster in *Plxnb2* mutants than in controls.

After completion of cerebellar development, *Plxnb2* mutant CGNs displayed strikingly disorganized parallel fibers (*Figure 5H*, *Videos 5* and *6*). Instead of being restricted to a thin sub-layer of the ML in controls, parallel fibers were more spread out in *Plxnb2* mutants. Some CGN axons were even completely misprojecting deep into the cerebellar white matter where they run along myelinated Purkinje cell axons and mossy fibers (*Figure 5F and F-II* and *Figure 5—figure supplement 2A*). The ectopic CGN axons keep some of their normal characteristics since they did not get myelinated (*Figure 5—figure supplement 2A,B*). In the IGL of *Plxnb2* mutants, the labeled CGNs were more dispersed, and some formed patches at the cerebellar surface (*Figure 5G and HIII*).

According to the 'stacking model', developing parallel fibers accumulate in the ML in an inside-out time sequence, with CGNs born later extending parallel fibers above those of earlier born CGNs (*Espinosa and Luo, 2008*). The dispersion of proliferating CGNs in the *Plxnb2* mutants EGL together with the presence of proliferation markers in tangentially migrating CGNS, suggest that their developmental clock is perturbed. Accordingly, *Plxnb2* mutant parallel fibers were much more scattered across the ML. To better visualize the spatiotemporal organization of parallel fibers in the ML, we electroporated a GFP-expressing vector at P7, followed by a td-Tomato-expression vector at P11. With this method, we could label two pools of early-born (GFP⁺) and late-born (Tomato⁺) CGNs in the same mouse (*Figure 6A*). As expected, GFP⁺ and Tomato⁺ parallel fibers were clearly segregated in control mice. Strikingly, in *Plxnb2* mutants, parallel fibers lost this inside-out organization in the ML (*Figure 6A*). At P11, early GFP⁺ fibers occupied a space within the ML twice larger than in controls and largely overlap with later-born Tomato⁺ parallel fibers, which were also more spread than in controls (*Figure 6A*).

We next explored the development of Purkinje cell dendritic arbors and their connections with parallel fibers. Immunohistochemistry of thick sagittal sections made of cerebella 48 hr after electroporation with GFP, showed that the distal tips of Purkinje cell dendrites reached until the newly

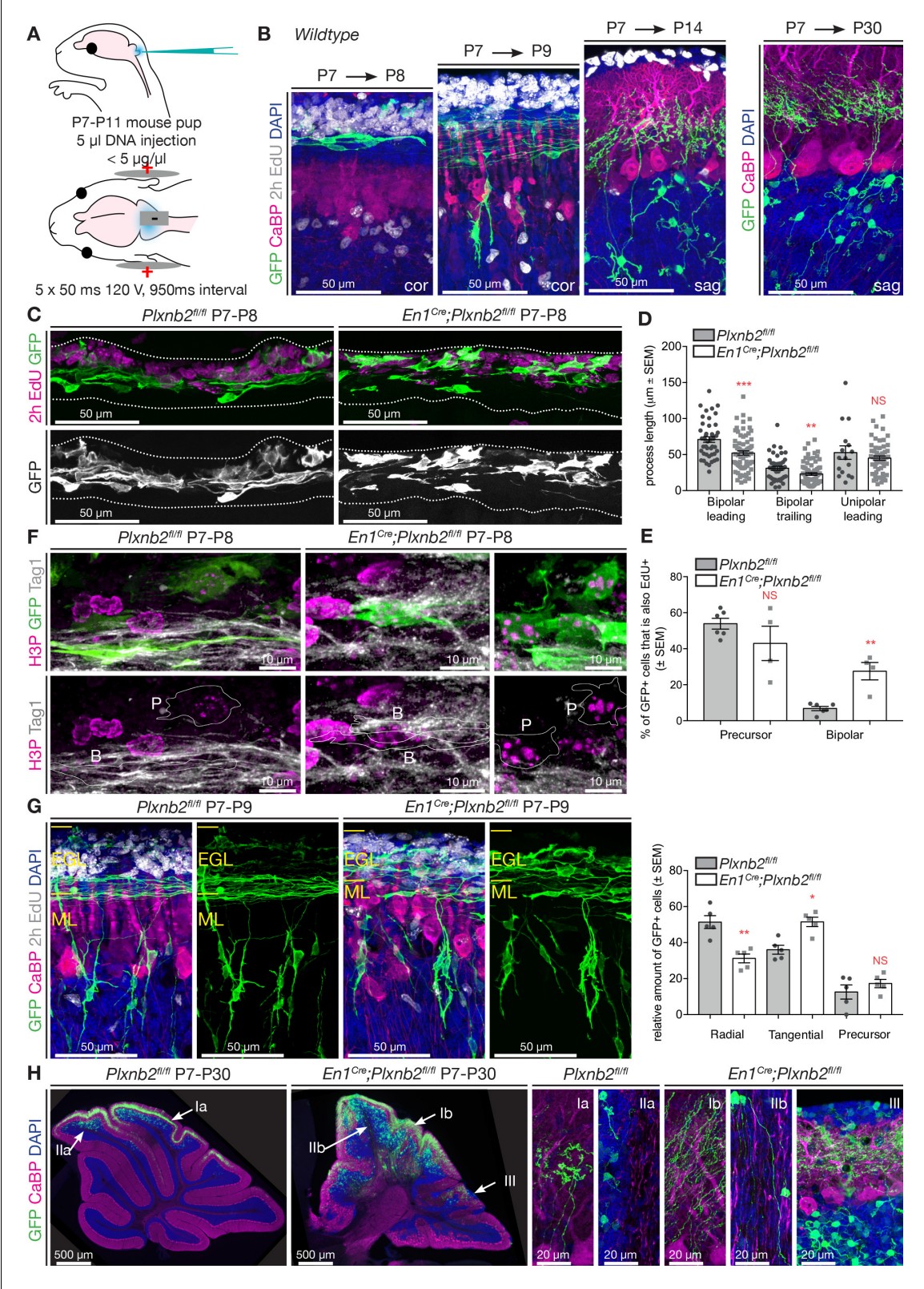

**Figure 5.** *Plxnb2* mutant CGNs display aberrant proliferative and tangential stages. (**A**) Schematic representation of the in vivo cerebellum electroporation protocol. See Materials and methods for details. (**B**) Cerebellar sections of electroporated brains at 1 day, 2 days, 1 week and 3 weeks after electroporation at P7, to illustrate the different stages of CGN development. Sections were immunostaining for GFP, Calbindin (CaBP), to label Purkinje cells. EdU was injected 2 hr prior to fixation, to label proliferating CGNs in the oEGL. One day after electroporation, GFP+ CGNs are still

*Figure 5 continued on next page*

*Figure 5 continued*

proliferating or became postmitotic and initiated tangential migration. Two days after electroporation GFP$^+$ CGNs start to migrate radially toward the IGL. One week after electroporation all GFP$^+$ CGNs reached the IGL, where they start growing dendrites. After 3 weeks, GFP$^+$ cells have their characteristic morphology with 3–4 claw-shaped dendrites. (C) Immunohistochemistry of coronal sections of cerebellum 1 day post-electroporation. GFP shows the electroporated CGNs and EdU, which was injected 2 hr before fixation, labels proliferating CGNs. Both the distribution and the morphology of migrating *Plxnb2* mutant GFP$^+$ CGNs are altered. (D) The graph shows aberrant process length of tangentially migrating CGNs in *En1$^{Cre}$;Plxnb2$^{fl/fl}$* pups. Error bars represent SEM. Bipolar leading process (longest process): ctl 70.86 ± 3.94 µm *vs.* mut 52.12 ± 2.92 µm, MWU(955) p=0.0002. Bipolar trailing process: ctl 31.1 ± 2.68 µm *vs.* mut 23.1 ± 1.84 µm, MWU(1117) p=0.0057. Unipolar leading process: ctl: 52.71 ± 9.32 µm *vs.* mut 45.07 ± 3.02 µm. MWU(416) p=0.75 (not significant). Forty-four wildtype bipolar cells and 73 mutant bipolar cells, and 16 wildtype unipolar and 66 mutant unipolar cells of 6 wildtype and 4 *Plxnb2* mutant animals were quantified. (*Figure 5—source data 1*) (E) Quantification of the % of EdU$^+$ and GFP$^+$ GCNs. In *Plxnb2* mutants, many bipolar GFP$^+$ GCNs are also EdU$^+$, unlike in controls (see *Figure 5—figure supplement 1*). By contrast the % EdU$^+$/GFP$^+$ GCN precursors is similar in *Plxnb2$^{fl/fl}$* controls and *En1$^{Cre}$;Plxnb2$^{fl/fl}$* mutants. A total of 447 ctl and 297 mutant precursors, and 451 ctl and 533 mutant bipolar CGNs were counted, from 6 wildtype and 4 *Plxnb2* mutant animals. Error bars represent SEM. Precursors: ctl 53.91 ± 3.01% *vs.* mut 42.97 ± 9.51%, MWU(8) p=0.48 (not significant). Bipolar cells: ctl 6.82 ± 1.17% *vs.* mut 27.53 ± 4.86%, MWU(0) p=0.0095 (*Figure 5—source data 1*). (F) P8 coronal sections of the cerebellum, 1 day post-electroporation. Mitotic CGNs in the EGL are immunostained with anti-H3P antibodies. At this stage, GFP$^+$ cells are either in a precursor state (outlined and marked P) or display a clear bipolar morphology (outlined and marked B) and express Tag1, a marker of tangentially migrating CGNs. In controls, only CGN precursor cells express H3P, whereas in *Plxnb2* mutants, H3P is found in precursors but also in some Tag1$^+$ bipolar CGNs. (G) Coronal sections of the cerebellum 2 days post-electroporation. GFP immunostaining labels the electroporated CGNs, and EdU (injected 2 hr before fixation) stains proliferating CGNs. Calbindin (CaBP) labels Purkinje cells. GFP$^+$ cells were counted and grouped in radial, tangential and precursor cell stages based on their morphology. In controls, most CGNs have reached radial stage 2 days after electroporation. By contrast, many GFP$^+$ CGNs are still in the tangential phase in *Plxnb2* mutants. Radial CGNs are not labeled by EdU. Graph shows that in *Plxnb2* mutants, more cells are in the radial stage (ctl 50 ± 2.77% *vs.* mut 38.28 ± 2.37%, MWU(1) p=0.0159) and less cells in the tangential stage (ctl 34 ± 1.33% *vs.* mut 47.85 ± 2.37%, MWU(0) p=0.0079). There is no significant difference in cells still in the precursor stage (ctl 16 ± 1.98% *vs.* mut 13.9 ± 1.71%. MWU(10) p=0.65). Error bars represent SEM. 899 ctl and 744 mutant CGNs were counted, from five animals per genotype (*Figure 5—source data 1*). (H) Sagittal sections of the cerebellum more than 3 weeks after electroporation with GFP. Electroporated CGNs are stained with GFP, Purkinje cells with Calbindin (CaBP) and sections were counterstained with DAPI. Three different types of defects are seen in *Plxnb2* mutants: (I) Parallel fibers that usually occupy a thin part within the molecular layer (Ia) disperse through the entire molecular layer in the mutant (Ib); (II) Whereas the white matter of control cerebella is devoid of parallel fibers (IIa), some mutant CGNs send their axons into the cerebellar white matter (IIb); and (III) ectopic patches of CGNs accumulate at the cerebellar. Ectopic CGNs still acquire their characteristic morphology. Scale bars: (B, C, E): 50 µm; D: 10 µm; (F) overview panels: 500 µm, high-magnification panels: 20 µm.

The online version of this article includes the following source data and figure supplement(s) for figure 5:

**Source data 1.** CGN morphology in vivo and colocalization with proliferation markers.
**Figure supplement 1.** Identity of electroporated cells and quantification of morphological features.
**Figure supplement 1—source data 1.** Identity of electroporated cells in vivo, morphology of electroporated CGNs.
**Figure supplement 2.** Misplaced and misprojecting CGNs keep their identity.

formed GFP$^+$ CGN processes (the forebears of parallel fibers) at the border of the EGL and ML (*Figure 6B*, *Figure 6—figure supplement 1A*). Vglut1$^+$ puncta, specific to CGN-Purkinje synapses, were distributed in a proximal-to-distal gradient: high at the trunk of the Purkinje dendritic tree, and low at the distal branches (*Figure 6—figure supplement 1A*). In P9 *Plxnb2* mutants, the young GFP$^+$ parallel fibers ran throughout the entire ML between the Purkinje cell dendrites, which in turn appeared disorganized and more branched than controls. Interestingly, in *Plxnb2* mutants, Vglut1$^+$ synapses extended to the tip of the Purkinje cell dendrites. We quantified the ratio between Vglut1$^+$ puncta (fluorescent integrated density) at the distal ends and the proximal base of the dendritic arbor, and found significantly more synapses on the distal end compared to control (*Figure 6—figure supplement 1A*). At later stages, no difference could be observed in the distribution of CGN-Purkinje synapses (*Figure 6—figure supplement 1B*).

We next compared the position of the nascent parallel fibers in the cerebellum of control and mutant mice electroporated at P7 (*Figure 6B*) or P11 (*Figure 6C*) and collected 2 days later. In controls, nascent parallel fibers extended at the bottom of the iEGL just above the tips of Purkinje dendrites *Plxnb2* mutant brains. Where at P9, the Purkinje cells in the *Plxnb2* mutant seem underdeveloped and developing parallel fibers cross the entire ML (*Figure 6B*). New parallel fibers crisscrossing deep into the ML were also found in P13 *Plxnb2* mutant, when the ML is much larger and the Purkinje dendrites are more developed (*Figure 6C*).

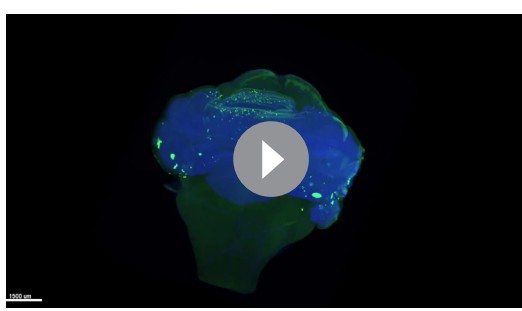

**Video 5.** 3D movie of P65 iDISCO+ cleared *Plxnb2^{fl/fl}* cerebellum electroporated at P7 with GFP. Whole mount immunostaining was performed with GFP to stain electroporated CGNs, FoxP2 to visualize Purkinje cell bodies, and TO-PRO-3 to stain all cell nuclei and visualize cerebellar anatomy.
https://elifesciences.org/articles/60554#video5

## Plxnb2 mutant CGNs display striking migration phenotype in vitro

To gain more insights into the behavior of *Plxnb2* mutant CGNs, we cultured EGL explants from 4- to 5-day-old pups. As previously described, CGNs exiting the explants follow a developmental sequence closely resembling in vivo CGNs (*Kawaji et al., 2004*; *Kerjan et al., 2005*). As in the oEGL, CGN precursors divide inside the explant or close to it (see below and *Yacubova and Komuro, 2002*). After plating, postmitotic cells become bipolar, grow long neurites and migrate away from the explant by nuclear translocation, as during tangential migration in the iEGL (*Figure 7A*). Two to 3 days later, CGNs start aggregating and form satellites around the explant (*Figure 7A*; *Kawaji et al., 2004*). Immunocytochemical analysis of young explants with Pax6, a marker for pre-and postmitotic CGNs, confirmed that the cells migrating away from the explant were CGNs (*Figure 7—figure supplement 1A*). Although the explant contained GFAP⁺ glial cells extending processes outward, their cell bodies seldom left the explant (*Figure 7—figure supplement 1B*).

We next compared explants from P4-5 *Plxnb2^{fl/fl}* or *En1^{Cre}*;*Plxnb2^{fl/fl}* EGL after 1 day in vitro (DIV) and noticed a difference in outward CGN migration. DAPI-stained nuclei from *Plxnb2* mutant CGNs stayed closer to the explant (*Figure 7B,C*). Furthermore, β-III tubulin staining revealed a difference in neurite outgrowth (*Figure 7B*), with shorter and more fasciculated neurites in *Plxnb2* mutants. To better analyze the morphology of individual CGNs, we labeled a subset of CGNs with GFP by ex vivo electroporation just prior to dissecting the cerebella for EGL cultures. Almost all GFP⁺ cells were positive for CGN markers such as Pax6 (*Figure 7—figure supplement 2A*) and Sema6A (*Figure 7—figure supplement 2B*), and did not resemble GFAP⁺ glial cells (*Figure 7—figure supplement 2C*). In controls, GFP⁺ CGNs migrating away from the explant at DIV1 either had a bipolar morphology, with ovoid cell bodies and long processes, or were more roundish cells without clear polarity and only short protrusions (*Figure 7D*). These multipolar cells, are probably CGN precursors as previously proposed (*Yacubova and Komuro, 2002*). Strikingly, at DIV1 in *Plxnb2* mutant explant cultures, the proportion of multipolar GFP⁺ CGNs was significantly increased (65.42 ± 2.37% in mut *vs.* 48.88 ± 2.65% in ctl, MWU(353) p<0.0001) and the proportion of bipolar cells was decreased (*Figure 7D,E*). However, by DIV2 almost all cells – control or mutant – had a bipolar morphology (*Figure 7D,E*). Interestingly, as observed in vivo (*Figure 5C*), DIV1 bipolar *Plxnb2* mutant CGNs had shorter processes than control cells (*Figure 7D,F*). Finally, bipolar cells could be further subdivided into two categories: cells that connected with their trailing process to the original explant, and cells that were disconnected from the explant. Interestingly, at DIV2, *Plxnb2* mutant had a higher proportion of GFP⁺ CGNs that were not attached to the explant (*Figure 7D,G*).

To better evaluate the consequence of *Plxnb2* deletion on the migration of bipolar CGNs, we next performed time-lapse video-microscopy of DIV1 and DIV2 EGL explant cultures.

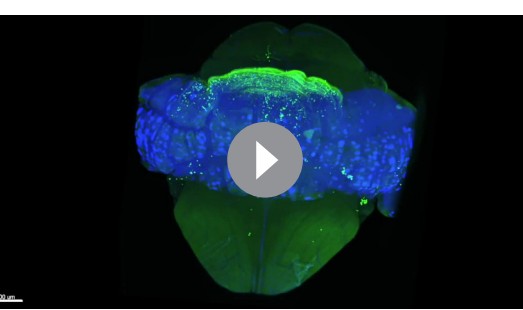

**Video 6.** 3D movie of P65 iDISCO+ cleared *En1^{Cre}*; *Plxnb2^{fl/fl}* cerebellum electroporated at P7 with GFP. Whole mount immunostaining was performed with GFP to stain electroporated CGNs, FoxP2 to visualize Purkinje cell bodies, and TO-PRO-3 to stain all cell nuclei and visualize cerebellar anatomy.
https://elifesciences.org/articles/60554#video6

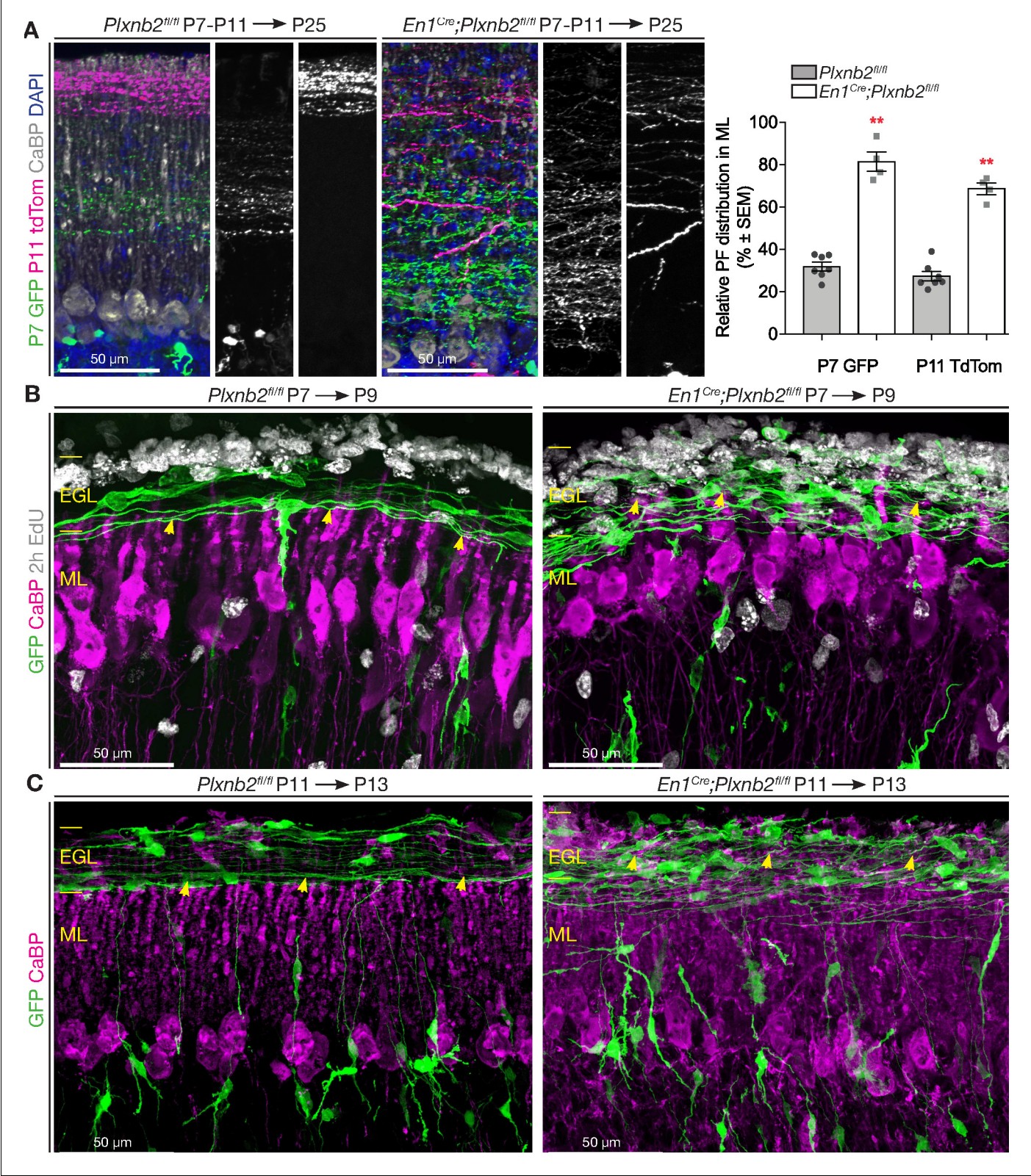

**Figure 6.** Abnormal parallel fiber layering in *Plxnb2* mutants. (**A**) Coronal sections of the cerebellum of P25 mice electroporated with GFP at P7 and re-electroporated with tdTomato (tdTom) at P11. Double immunostaining for GFP and tdTomato. In control (left) the parallel fibers of CGNs that became postmitotic early (GFP⁺) are at the bottom of the molecular layer, whereas the CGNs that became postmitotic later (tdTom⁺) extend parallel fibers at the surface of the molecular layer. In *En1^Cre^;Plxnb2^fl/fl^* mutants, there is an important overlap in the molecular layer, between parallel fibers of early and

*Figure 6 continued on next page*

*Figure 6 continued*

late-born CGNs. The graph shows a quantification of the portion of the molecular layer that is occupied by parallel fibers of either early (GFP⁺) or late (tdTom⁺) CGNs (eg. (GFP⁺ width / ML total width) x 100%). Error bars represent SEM. The molecular layer measurements and its double-electroporated parallel fiber content was averaged from three different points per cerebellum from 7 *Plxnb2^fl/fl* and 4 *En1^Cre;Plxnb2^fl/fl* cerebella. P7 GFP ctl: 31.96 ± 2.07% *vs.* mut: 81.48 ± 4.53% (MWU(0) p=0.0061) and P11 tdTom ctl: 27.45 ± 2.26% *vs.* mut: 68.74 ± 2.75% (MWU(0) p=0.0061). (**Figure 6—source data 1**) (B) Coronal sections of cerebella electroporated at P7 and collected at P9 (EdU was injected 2 hr before termination). Sections were stained for GFP, CaBP, and EdU. In controls (left panel), nascent parallel fibers normally extend at the base of the iEGL, just above the tips of developing Purkinje dendritic arbors (yellow arrowheads). However, in *Plxnb2* mutant (right panel) parallel fibers extend throughout the EGL and cross the Purkinje dendrites in the ML (yellow arrowheads indicate the tips of Purkinje dendrites). (C) The abnormal presence of young GFP⁺ parallel fibers deep in the molecular layer is also seen on coronal sections of cerebella electroporated at P11 and collected at P13 (Control, left panel and *Plxnb2* mutant, right panel). Scale bars 50 μm.

The online version of this article includes the following source data and figure supplement(s) for figure 6:

**Source data 1.** Parallel fiber distribution.
**Figure supplement 1.** Abnormal localization of parallel fiber synapses in *Plxnb2* mutant.
**Figure supplement 1—source data 1.** Synaptogenesis between parallel fibers and Purkinje cells.

Interestingly, whereas control GFP⁺ CGNs usually migrated away from the explant in a straight and radial direction, *Plxnb2* mutant GFP⁺ CGNs sometimes reversed direction one or even multiple times during the acquisition period (**Figure 8A**, **Video 7**). The afore-mentioned difference in CGN process lengths during tangential migration could also be observed in Videos. Although the speed of migrating bipolar CGNs was not changed (**Figure 8B**), both the relative amount of distance and time going in negative direction (moving back toward the explant) were significantly increased in *Plxnb2* mutant CGNs (**Figure 8C**). These results probably explain why in fixed DIV1 cultures, *Plxnb2* mutant CGN nuclei appear closer to the explant (**Figure 7B**).

Taken together, both our in vivo and in vitro data support an abnormal outgrowth of processes in *Plxnb2*-deficient tangentially migrating CGNs.

## Plxnb2 mutant CGN precursors show aberrant proliferation and movement

Since we observed slight differences in cell-cycle completion and an aberrant localization of proliferating precursors in EGL sections (**Figure 4**), we also aimed at analyzing proliferation in EGL explant cultures. EdU was added to the medium 2 hr before fixation (**Figure 9A**). Although there was a much lower amount of DAPI nuclei visible around the explant (**Figure 9A,B**), a similar amount of EdU⁺ nuclei was observed (**Figure 9A,C**). In addition, the explants did not show a difference in EdU⁺ nuclei that also stained for H3P (**Figure 9A,D**).

With longer application of EdU, there was still no difference in the distribution of EdU between multipolar and bipolar cells at DIV1 (**Figure 9E,F**). Nevertheless, a larger portion of multipolar cells was positive for EdU (**Figure 9F**), suggesting that these multipolar cells actually were CGN precursors that escaped from the explant. Interestingly, at DIV2, many more cells with a bipolar appearance had an EdU⁺ nucleus in *Plxnb2* mutant explants (**Figure 9E,G**). Therefore, these data suggest that in mutant explants, bipolar cells are still generated long after the explants are seeded, suggesting that the in vitro proliferation of CGN precursors is differentially phased compared to controls.

Intrigued by the potential precursor properties of the multipolar GFP⁺ CGNs in the cultures, we attempted to follow their behavior in our time-lapse recordings. As evident from the fixed cultures, the proportion of multipolar cells at the beginning of the time-lapse acquisition period (at DIV1) was twice as big in mutant explants compared to control (**Figure 10—figure supplement 1B**). At the end of the time-lapse acquisition period (around DIV2) almost all control cells reached a bipolar state, whereas in *Plxnb2* mutant explants a large portion still appeared multipolar (**Figure 10—figure supplement 1B**). The time-lapse acquisitions of multipolar CGNs confirmed their ability to proliferate. They divided and produced two daughter cells that became bipolar and henceforth commenced their tangential migration phase (**Figure 10A**, **Video 8**). This confirms that multipolar cells are probably CGN precursors. Before they divided, *Plxnb2* mutant multipolar CGNs showed a striking increase of movement compared to controls (**Figure 10B,C**). The presence of at least twice as much multipolar CGNs moving around mutant explants compared to controls (**Figure 10—figure supplement 1**), probably explained why more multipolar cell divisions per explant were counted

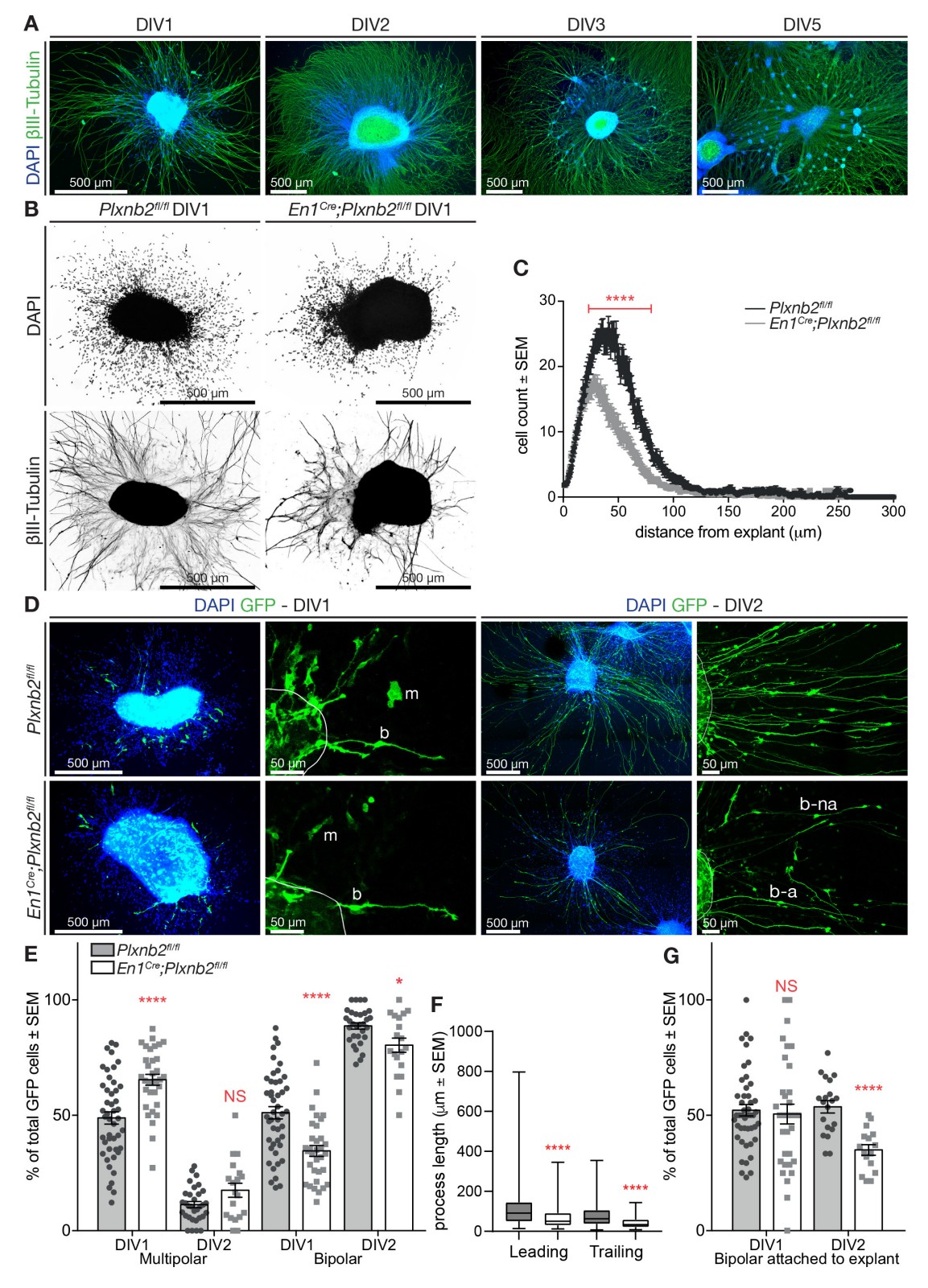

**Figure 7.** *Plxnb2* CGNs recapitulate in vitro the developmental defects found in vivo. (**A**) EGL explants from P4-P5 wildtype cerebella, fixed after 1, 2, 3, and 5 days in vitro (DIV). Immunocytochemistry for βIII-tubulin and DAPI shows that cells migrate away from the explant and extend long neurites. After DIV2, cells start accumulating in clusters around the original explant. (**B**) EGL explants from P4-P5 *Plxnb2^fl/fl* and *En1^Cre;Plxnb2^fl/fl* cerebella at DIV1. Cultures were stained for DAPI and βIII-tubulin. *Plxnb2* mutant explants show DAPI⁺ nuclei closer to the explant and different neurite outgrowth. (**C**)

*Figure 7 continued on next page*

*Figure 7 continued*

DAPI$^+$ nuclei around the explant were counted and their migration was assessed using a Sholl-analysis. Graph shows that less cells migrate from *En1$^{Cre}$*; *Plxnb2$^{fl/fl}$* explants and that they stay closer to the explant. Multiple t-test with the Holm-Sidak method were applied to the mean intersections of DAPI$^+$ nuclei with the Sholl circles. p<0.0001. A total of 36 controls and 34 mutant explants were analyzed from three different experiments. Error bars represent SEM. (*Figure 7—source data 1*) (D) EGL explants from cerebella electroporated ex vivo with GFP and fixed at DIV1 and DIV2. Immunocytochemistry for GFP and DAPI shows the morphology of migrating cells. GFP$^+$ CGNs have either a multipolar (**m**) or a bipolar (**b**) shape. After DIV2, almost all GFP$^+$ cells have a bipolar morphology, with their trailing process attached (**b–a**) or not (**b–na**) to the explant. (E) Quantification of the proportion of GFP$^+$ CGNs with multipolar or bipolar morphologies. Data is expressed as percentage from total number of GFP$^+$ cells per explant ± SEM. DIV1 multipolar: ctl 48.88 ± 2.65% *vs.* mut 65.42 ± 2.37%, MWU(353) p<0.0001. DIV1 bipolar: ctl 51.12 ± 2.65% *vs.* mut 34.58 ± 2.37%, MWU(353) p<0.0001. DIV2 multipolar ctl 11.36 ± 1.33% *vs.* mut 17.58 ± 3.02%, MWU(226) p=0.13, not significant. DIV2 bipolar ctl 88.67 ± 1.37% *vs.* mut 80.31 ± 3.07%, MWU(189) p=0.03. All GFP$^+$ cells (amounts between brackets) were counted from 46 ctl (2728) and 33 mut (835) explants (DIV1) and 32 ctl (2284) and 19 mut (617) explants (DIV2) from at least three different experimental repeats. (*Figure 7—source data 1*) (F) Quantification (Box-plots) of leading and trailing process length of bipolar GFP$^+$ CGNs at DIV1. Leading ctl 106 ± 3.9 μm *vs.* mut 67.6 ± 5.62 μm, MWU(11147) p<0.0001; trailing ctl 79.5 ± 3.4 μm *vs.* mut 40.9 ± 2.76 μm, MWU(68833) p<0.0001. A total of 385 ctl and 93 mut cells were analyzed from 29 ctl and 13 mut explants from three experimental repeats (*Figure 7—source data 1*). (G) Proportion of bipolar GFP$^+$ CGNs attached to the explant. DIV1 attached: ctl 52.19 ± 2.49% *vs.* mut 50.55 ± 4.27%, (MWU(709.5)) p=0.63, not significant. DIV2 attached: ctl 53.63 ± 2.69% *vs.* mut 34.92 ± 2.31%, MWU(36.5) p<0.0001. All GFP$^+$ cells (amounts between brackets) were counted from 46 ctl (2728) and 33 mut (835) explants (DIV1) and 32 ctl (2284) and 19 mut (617) explants (DIV2) from at least three different experimental repeats (*Figure 7—source data 1*). Scale bars: overviews 500 μm (A, B, D); magnifications in (D): 50 μm.

The online version of this article includes the following source data and figure supplement(s) for figure 7:

**Source data 1.** EGL explants: in vitro CGN morphology.
**Figure supplement 1.** Migrating cells have a CGN identity.
**Figure supplement 1—source data 1.** Identity of electroporated cells migrating out of EGL explants.
**Figure supplement 2.** Electroporated cells migrating away from EGL explants have CGN identity.
**Figure supplement 2—source data 1.** Identity of cells migrating out of EGL explants.

---

throughout the acquisition period (*Figure 10D*). We never observed more than one division of a single multipolar cell in our acquisitions, and whenever visible, all daughter cells eventually adopted a bipolar shape and started migration. However, we found that the time taken by the daughter cells to become bipolar after cytokinesis was increased in mutants compared to controls (*Figure 10E*). During this in-between period, the daughter cells were again very motile and they appeared to struggle to become polarized (*Video 8*).

## Discussion

### Revisiting Plexin-B2 function in cerebellum development at a cellular level

Cerebellar granule cells are one of the best models to study neuronal migration as they display a large palette of migratory behavior at embryonic and postnatal stages (*Chédotal, 2010*). Our work confirms that the expression pattern and function of the Plexin-B2 receptor in CGN development, is quite unique. Plexin-B2 is only expressed in proliferating CGN precursors and silenced as soon as CGNs enter the iEGL and initiate their migration. Previous studies have shown that molecular layer organization is severely perturbed in Plexin-B2 knockouts (*Deng et al., 2007*; *Friedel et al., 2007*). Here, we used two distinct Cre lines (*En1$^{Cre}$* and *Wnt1-Cre*) to silence *Plxnb2* function in the EGL and show that they fully phenocopy the null allele but have a normal viability. This, together with similar observation made with the *Atoh1cre* line (*Worzfeld et al., 2014*) and our time-lapse studies in EGL explants, shows that *Plxnb2* acts cell autonomously in cerebellar CGN precursors. However, the consequence of Plexin-B2 deficiency at cellular and subcellular levels were unknown as the extremely high number and density of cerebellar CGNs, as well as their molecular and genetic homogeneity do not facilitate the in situ analysis of the evolution of their morphology during development. Here, we show that the use of a tripolar electrode is an optimal method to express transgenes in postnatal CGNs. Mosaic analysis with double markers (MDAM) can reveal the individual morphology of developing CGNs but it requires specific lines and complex genetic crosses (*Zong et al., 2005*). Viral vectors have been used to express fluorescent proteins in developing CGNs but the delay between the infection and the transgene expression does not allow to observe the early phases of CGN development in the EGL (*Park et al., 2019*). The size of the transgene is

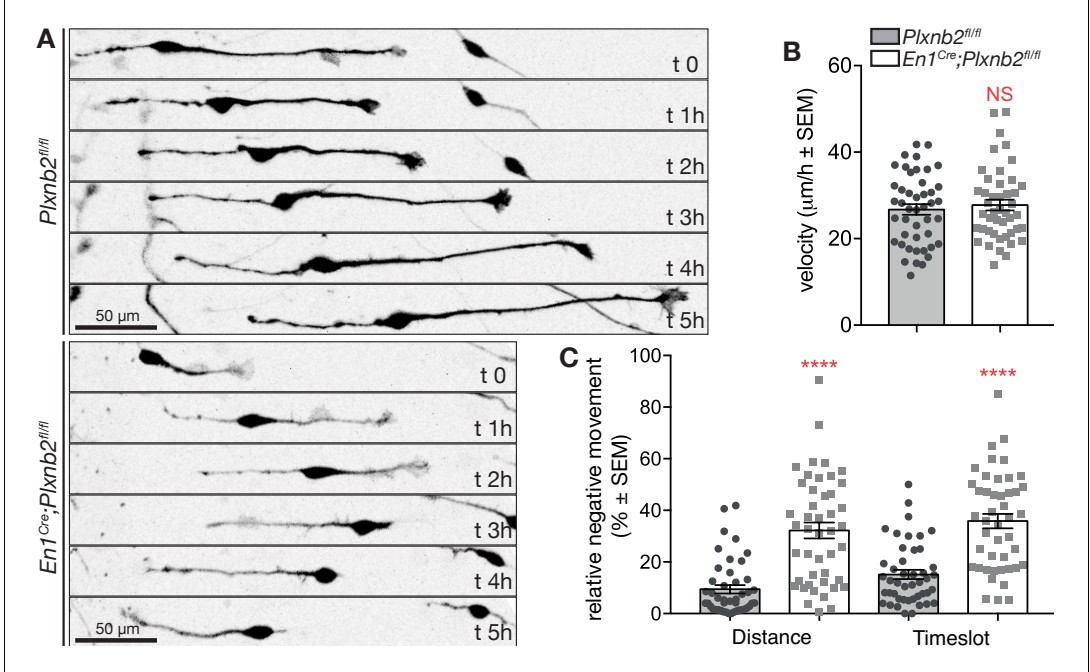

Figure 8. *Plxnb2* mutant CGNs in culture display aberrant tangential migration. (**A**) 15-minute-time-lapse confocal still images at t 0, 1, 2, 3, 4, and 5 hr showing GFP$^+$ CGNs migrating from a DIV1 explant (located on the left side of the images). Scale bars 50 µm. (**B**) Bipolar CGNs migrate at an equal speed. (ctl: 26.75 ± 1.23 µm/h *vs.* mut: 27.77 ± 1.25 µm/h, MWU(973) p=0.75, not significant). Forty-five bipolar cells were tracked for each condition, from 13 ctl and 11 mut cultures from five independent experiments. Error bars represent SEM. (**C**) *En1$^{Cre}$;Plxnb2$^{fl/fl}$* CGNs cover more distance (ctl 9.42 ± 1.60% *vs.* mut 32.21 ± 3.10%, MWU(306) p<0.0001) and spend more time (ctl 15.15 ± 1.77% *vs.* mut 35.86 ± 2.80%, MWU(352.5) p<0.0001) going in negative direction (toward instead of away from the explant). Forty-five bipolar cells were tracked for each condition, from 13 ctl and 11 mut cultures. Error bars represent SEM. (*Figure 8—source data 1*).

The online version of this article includes the following source data for figure 8:

**Source data 1.** EGL explants: live imaging of bipolar CGN migration.

also limited. In rodents, CGNs are produced postnatally and the superficial location of the EGL under the skull makes it easily accessible. Therefore, direct electroporation of plasmids into the cerebellum using tweezer electrodes has been performed to target developing CGNs ex vivo (*Govek et al., 2018*; *Renaud and Chédotal, 2014*) or in vivo (*Konishi et al., 2004*; *Umeshima et al., 2007*). Here, we have successfully adapted a triple electrode method, previously designed to target ventricular zone progenitors in the embryonic cerebellum (*dal Maschio et al., 2012*), to express fluorescent proteins in a large domain of the postnatal EGL covering multiple folia. The methods have been used between birth and at least P11 with comparable outcome. Importantly, we show that it allows multiple rounds of electroporation at different time-points, which allows to study parallel fiber layering in the molecular layer without viral vectors or MADM lines.

The analysis of CGN morphology with GFP electroporation, showed that in absence of

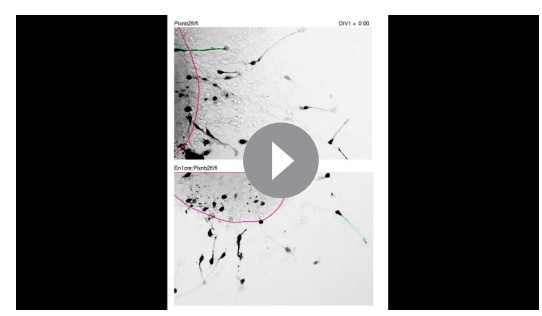

**Video 7.** Representative examples of confocal time-lapse recording of EGL explant cultures of P4-P5 *Plxnb2$^{fl/fl}$* and *En1$^{Cre}$;Plxnb2$^{fl/fl}$* cerebella with 15 min interval, starting from DIV1. Cerebella were electroporated ex vivo with GFP to visualize individual CGNs and follow their migration over time (some striking examples are pseudo-colored). Control CGNs with a bipolar morphology migrate away from the explant in a straight direction. *Plxnb2* mutant CGNs change their direction of migration multiple times and cover long distances in reverse direction (back to the explant).

https://elifesciences.org/articles/60554#video7

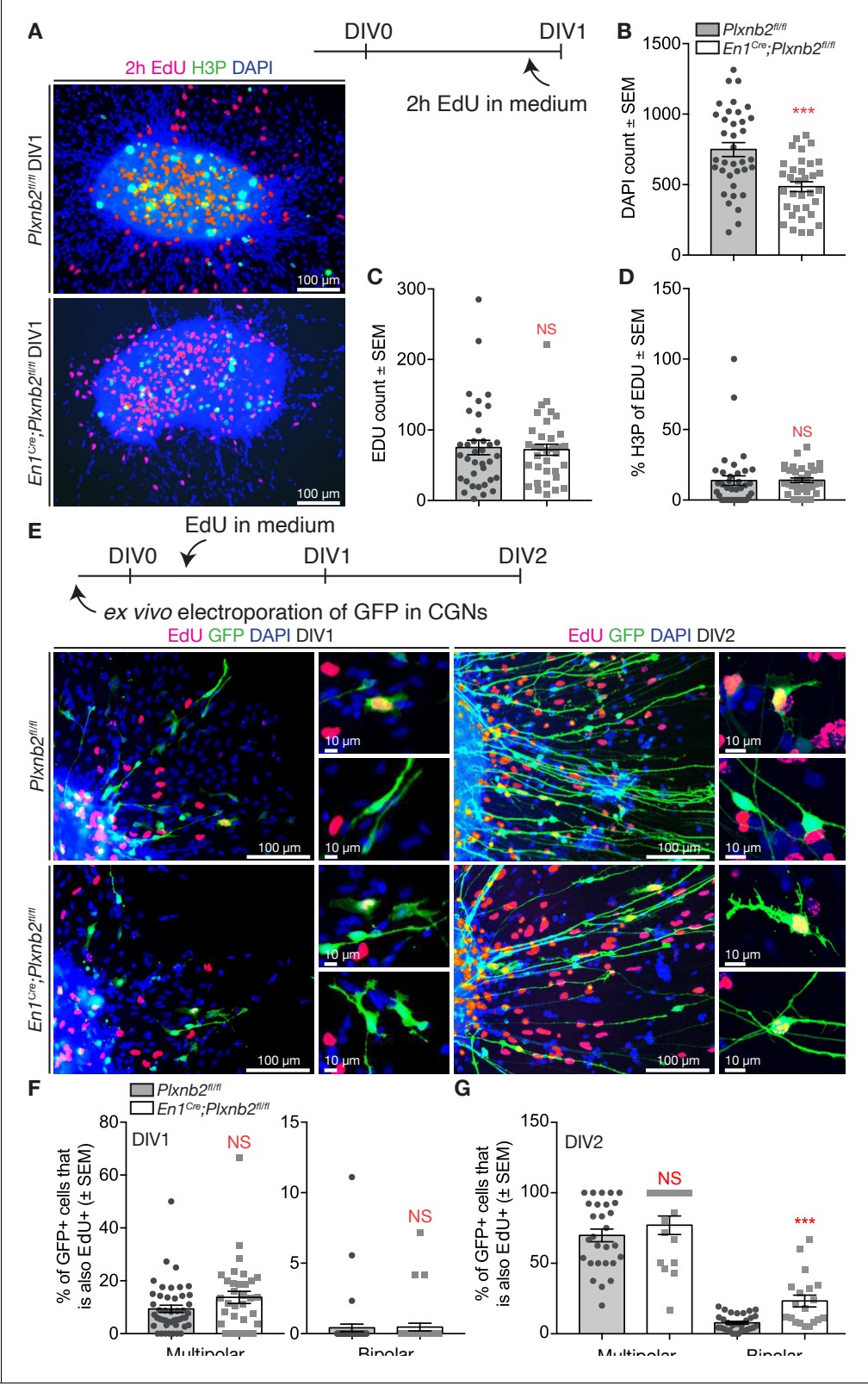

**Figure 9.** Aberrant proliferation of CGN precursors in *Plxnb2* mutant explants. (**A**) EGL explants from P4-P5 cerebella at DIV1. Two hr prior to fixation, 10 µM EdU was added to the culture medium. Cultures were stained for EdU, H3P, and DAPI. (**B**) The number of DAPI⁺ nuclei/migrating cells around DIV1 explants, is significantly decreased in *Plxnb2* mutants (485.79 ± 34.77 cells) compared to controls (748.89 ± 53.54 cells; MWU(290.5) p=0.0001). Error bars represent SEM. 36 ctl and 34 mut explants were analyzed from three different experiments. (***Figure 9—source data 1***) (**C**) At DIV1, there is

*Figure 9 continued on next page*

*Figure 9 continued*

no significant difference in the total amount of EdU$^+$ cells (that incorporated EdU in the last 2 hr) per explant. Ctl 75.19 ± 11.28 *vs.* mut 72.03 ± 7.85 cells (MWU(596.5) p=0.86. Error bars represent SEM. Thirty-six ctl and 34 mut explants were analyzed from three different experimental replicates (*Figure 9—source data 1*). (D) Likewise, the portion of EdU$^+$ cells also positive for H3P (an M-phase marker) at the moment of fixation) is similar in controls (13.77 ± 3.80%) and mutants (14.10 ± 0.93%, MWU(477) p=0.11). Error bars represent SEM. Thirty-six ctl and 34 mut explants were analyzed from three different experimental repeats (*Figure 9—source data 1*). (E) EGL explants from P4-P5 cerebella electroporated with GFP ex vivo. Ten µM EdU was added to the medium after 6 hr of culture. Explants were fixed at DIV1 or DIV2 and EdU incorporation was quantified in multipolar and bipolar GFP$^+$ cells. (F) Quantification of the proportion of multipolar and bipolar GFP$^+$ CGNs that have taken up EdU in the past 18 hr at DIV1 (EdU administered from 6 to 24 hr after plating). Multipolar ctl: 9.394 ± 1.35% *vs.* mut 13.75 ± 2.31%, MWU(595) p=0.10, not significant; bipolar ctl: 0.41 ± 0.27% *vs.* mut 0.47 ± 0.27%, MWU(740) p=0.78, not significant. Error bars represent SEM. 2814 ctl and 890 mut GFP$^+$ CGNs were counted from 47 ctl and 33 mut explants from three experimental replicates (*Figure 9—source data 1*). (G) Quantification of the proportion of multipolar and bipolar GFP$^+$ CGNs that have taken up EdU in the past 42 hr at DIV2 (EdU administered from 6 to 48 hr after plating). Multipolar ctl: 69.81 ± 4.50% *vs.* mut 77.03 ± 6.55%, MWU(189) p=0.25, not significant. Bipolar ctl: 7.76 ± 1.00% *vs.* mut 23.3 ± 4.20%, MWU(113) p=0.0001. Error bars represent SEM. 2284 ctl and 617 mut GFP$^+$ cells were counted from 32 ctl and 20 mut explants from three experimental repeats (*Figure 9—source data 1*). Scale bars 100 µm, high magnifications 10 µm.

The online version of this article includes the following source data for figure 9:

**Source data 1.** EGL explants: in vitro proliferation.

Plexin-B2, CGNs still follow the normal sequence of differentiation that in controls (*Komuro et al., 2001*; *Renaud and Chédotal, 2014*). They become bipolar, migrate tangentially, then tripolar and migrate radially across the molecular layer leaving behind parallel fibers. They also extend three to four dendrites undistinguishable from controls. However, it also shows that their parallel fibers are not properly layered and that some CGN axons are lost in the white matter. Interestingly, our results suggest that the mislocalized CGN axons remain unmyelinated in agreement with previous studies showing that axons have a unique profile of myelination (*Tomassy et al., 2014*).

## Plexin-B2 controls the timing of cell division in the EGL

Our results show that the size of the cerebellum is only slightly reduced in *En1$^{Cre}$;Plxnb2$^{fl/fl}$* mice thereby indicating that the generation of cerebellar neurons is almost unaffected by the lack of Plexin-B2. In addition, a significant fraction of tangentially migrating CGNs are still mitotically active in the EGL indicating that CGN precursors initiated differentiation before the completion of cell division. Interestingly, we also found that in EGL explant cultures the number of mitotically active CGNs with multipolar morphology is three times higher in *Plxnb2* mutant. Moreover, the time taken by the daughter cells to become bipolar after cytokinesis is increased in mutants compared to controls. This suggests that *Plxnb2* mutant CGNs might be maintained for a longer time in a multipolar and proliferating state, and that their ability to perform their final division could be altered, although they ultimately divide and produce a close to normal number of daughter cells. These results support a role for Plexin-B2 in cell division as previously described in cancer cell lines (*Gurrapu et al., 2018*) and in the developing kidney (*Xia et al., 2015*) where Plexin-B2 controls the orientation of the mitotic spindle. Interestingly, several studies suggest that plexins could control abscission, the final step of cell division, by promoting cytoskeleton disassembly at the intercellular bridge linking the two daughter cells. MICALs (molecule interacting with CasL) are oxidoreductases which regulate actin depolymerization and act directly (*Van Battum et al., 2014*; *Terman et al., 2002*) or indirectly (*Ayoob et al., 2006*; *Orr et al., 2017*) downstream of plexins (*Pasterkamp, 2012*; *Seiradake et al., 2016*). Interestingly, MICALs have been shown to control F-actin clearance during abscission (*Frémont et al., 2017*). Likewise, LARG, which associates with B-type plexins (*Pascoe et al., 2015*) is required for abscission in Hela cells (*Martz et al., 2013*). Although hypothetical, an involvement of Plexin-B2 in cytokinesis is also supported by a recent proteomic study which identified Plexin-B2 as one of the 489 proteins constituting the midbody, the large protein complex at the center of the intercellular bridge linking dividing cells (*Addi et al., 2020*). Of note, patients with mutations in citron kinase, a key component of the abscission machinery, display a severe disorganization of cerebellar cortex including the ectopic patches of CGNs observed in *Plxnb2* mutants (*Harding et al., 2016*; *Li et al., 2016*). Together, these results suggest that Plexin-B2 might control cell division in the outer EGL, a process which is also essential for orchestrating cerebellar foliation (*Legué et al., 2015*; *Otero et al., 2014*).

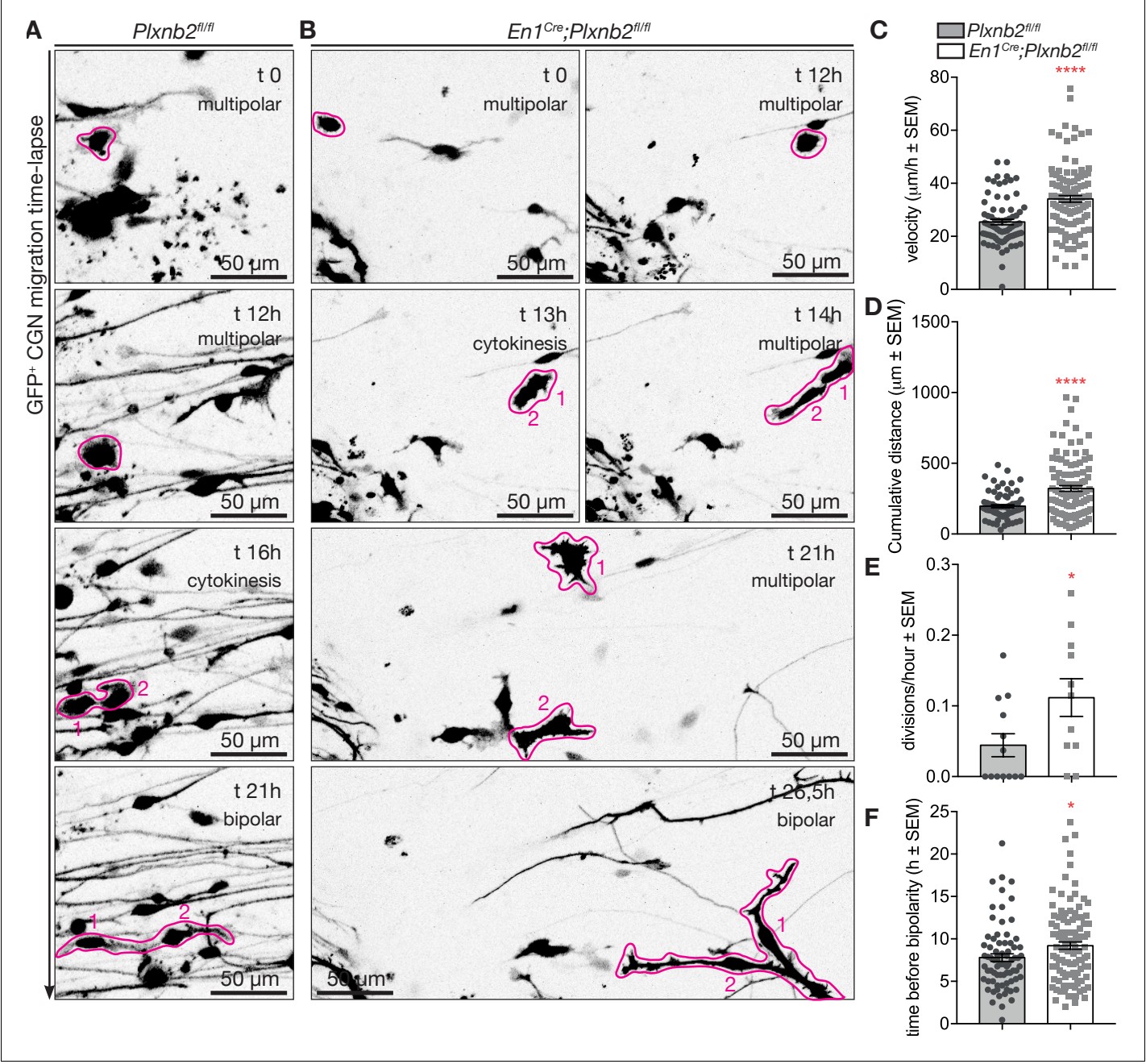

**Figure 10.** Aberrant CGN (precursor) motility before and after division in *Plxnb2* mutant explants. (A, B) Time-lapse confocal imaging series (21 hr) of GFP⁺ multipolar CGNs in control (A) and *Plxnb2* mutant explants at DIV1. (A) In a control, a multipolar cell (outlined in pink at t 0 hr) divides (cytokinesis, t 16 hr) to give rise to two daughter cells (1 and 2) which later adopt a bipolar morphology. (B) In a *Plxnb2* mutant, multipolar cells (outlined in pink) are more motile and the transition to the bipolar stage is delayed. Scale bars 50 μm. (C–F) Quantifications of multipolar cell velocity, cumulative distance before cytokinesis, time that daughter cells take to become bipolar after cytokinesis, and the amount of visible divisions of GFP⁺ cells per hour. Error bars represent SEM. 75 ctl and 107 mut multipolar GFP⁺ CGNs were tracked from 13 ctl and 11 mutant explants from five different experimental repeats. (C) Velocity ctl 25.41 ± 1.04 μm/h *vs.* mut 34.11 ± 1.24 μm/h, MWU(2279) p<0.0001 (*Figure 10—source data 1*). (D) Cumulative distance ctl 196.3 ± 11.08 μm *vs.* mut 321.8 ± 19.73 μm, MWU(2516) p<0.0001 (*Figure 10—source data 1*). (E) Time before bipolarity ctl 7.80 ± 0.45 hr *vs.* 9.21 ± 0.44 hr, MWU(3254) p=0.0298 (*Figure 10—source data 1*). (F) Divisions per hour ctl 0.044 ± 0.016 *vs.* mut 0.11 ± 0.027, MWU(36) p=0.034. (*Figure 10—source data 1*).

The online version of this article includes the following source data and figure supplement(s) for figure 10:

**Source data 1.** EGL explants: live imaging of multipolar CGNs.
**Figure supplement 1.** Quantification of the distribution of multi- and bipolar CGNs during time-lapse.

*Figure 10 continued on next page*

Figure 10 continued

**Figure supplement 1—source data 1.** Morphology of CGNs during live imaging.

## Plexin-B2 controls CGN migration

Our present study also shows that Plexin-B2 influences the migration of cerebellar CGNs. The overall distance reached by CGNs in EGL explants cultures is reduced in *Plxnb2* mutants, as previously described (*Maier et al., 2011*). Although delayed cell division probably contributes to this defect, it cannot be explained by a slower tangential migration, as our time-lapse analysis rather indicates that in *Plxnb2* mutants, multipolar CGNs are more motile, and cover twice as much cumulative distance than in controls. Moreover, in *Plxnb2* mutants, tangentially migrating bipolar CGNs alternate between forward (away from the explant) and rearward direction while control CGNs essentially migrate forward in this culture setup. The significant increase of multipolar and mitotically active CGNs, migrating around the explants suggest that CGN precursors become more motile without Plexin-B2.

Our data also provide evidence for altered CGN migration in vivo. The combination of GFP electroporation and EdU labeling shows that in *Plxnb2* mutants, CGNs remain for a longer time in tangential migration and that they take longer to initiate their radial migration. Moreover, tangentially migrating CGNs mix with CGN precursors and a significant fraction divides during tangential migration. These observations are in good agreement with previous studies which reported enhanced motility of *Plxnb2*$^{-/-}$ macrophages (*Roney et al., 2011*) and neuroblasts in the rostral migratory stream (*Saha et al., 2012*). Sema4D and Plexin-B2 were also reported to function as motogens for newborn cortical neurons (*Hirschberg et al., 2010*). A recent study also linked Plexin-B2 to microglial cell motility in the injured spinal cord, albeit negatively (*Zhou et al., 2020*). Together, these results show that in many developing tissue, Plexin-B2 is a key regulator of cell migration decisions.

## What could be the ligands and downstream partners mediating Plexin-B2 function in CGNs?

Our results confirm the essential and unique function of Plexin-B2 in granule cell development but the underlying molecular mechanisms remains an enigma. At least five of the Class four semaphorins (Sema4A, 4B, 4C, 4D and 4G) bind to Plexin-B2 (*Deng et al., 2007*; *Hirschberg et al., 2010*; *Maier et al., 2011*; *Tamagnone et al., 1999*; *Xia et al., 2015*; *Yukawa et al., 2010*). However, knocking down, *Sema4C* and *Sema4G*, the two class four semaphorins expressed in the developing cerebellar cortex, results in a mild phenotype (*Friedel et al., 2007*; *Maier et al., 2011*). This suggests that additional semaphorins could act redundantly or that other Plexin-B2 ligands could be involved. Angiogenin was recently shown to bind and signal through Plexin-B2 ligand in various cell types, but angiogenin does not activate the same pathways as class four semaphorins downstream of Plexin-B2 (*Yu et al., 2017*). Therefore, and although its expression in the developing cerebellum is unknown, angiogenin is unlikely to mediate Plexin-B2 function in the EGL. In addition, a spontaneous monkey mutant of angiogenin, does not display cerebellum defects (*Zhang and Zhang, 2003*).

Elegant genetic studies showed that the GAP and RBD domains of Plexin-B2, which mediate semaphorin activity, are essential for Plexin-B2 function in developing CGNs, but that the PDZ-binding domain is dispensable (*Worzfeld et al., 2014*). In vitro experiments suggested that the RBD domain of B-type plexins regulates their activity by interacting with Rho family small GTPases such as Ras, Rac1, Rnd1-3, and Rap1 (*Oinuma et al., 2004*; *Rohm et al., 2000*; *Tong et al., 2007*; *Turner et al., 2004*; *Vikis et al., 2000*; *Wang et al., 2012*; *Wang et al., 2013*; *Wylie et al., 2017*; *Zanata et al., 2002*).

However, structural biology studies showed that B-type plexins do not interact with M-Ras/R-Ras (*Wang et al., 2012*; *Wang et al., 2013*) and accordingly, in vivo evidence indicate that CGN developmental defects in *Plxnb2* mutants do not involve M-Ras/R-Ras (*Worzfeld et al., 2014*). Rac1 and to a lesser extent Rac3 are expressed in the postnatal EGL (*Nakamura et al., 2017*), but although their simultaneous inactivation perturbs CGN development, they primarily act on neuritogenesis and tangential migration of CGN precursors in the embryo, unlike Plexin-B2. Plexin-B2 interacts preferentially with Rnd3 (*Azzarelli et al., 2014*; *McColl et al., 2016*; *Wylie et al., 2017*) and in radially migrating cortical neurons, Plexin-B2 and Rnd3 have antagonistic function (*Azzarelli et al., 2014*).

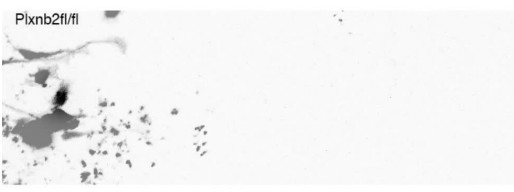

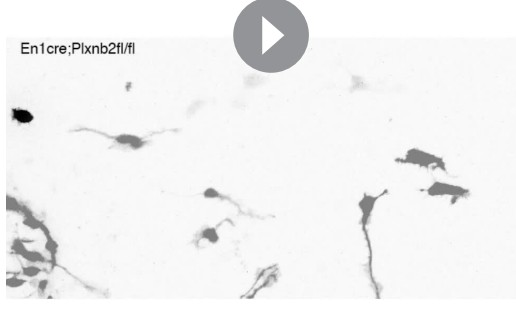

**Video 8.** Representative examples of confocal time-lapse recording of EGL explant cultures of P4-P5 *Plxnb2*<sup>fl/fl</sup> and *En1*<sup>Cre</sup>;*Plxnb2*<sup>fl/fl</sup> cerebella with 15-min interval, starting from DIV1. Cerebella were electroporated ex vivo with GFP to visualize individual CGNs and follow their migration over time.
https://elifesciences.org/articles/60554#video8

Although Rnd3 is expressed in EGL (*Ballester-Lurbe et al., 2009*), the structure of the cerebellum is normal in *Rnd3* knockout mice (*Mocholí et al., 2011*) (and data not shown). Interestingly, mammalian plexins have a higher GAP activity for Rap1 GTPases (*Wang et al., 2012*) and Plexin-B2/Rap1 interaction does not required Rnd3 (*McColl et al., 2016*). Therefore, Rap1 GTPases could be the main Plexin-B2 partners in developing CGNs, regulating the transition of CGN precursors from the oEGL to the iEGL. Accordingly, Rap1A/Rap1B are required for the transition of newborn cortical neurons from a multipolar to a bipolar state and their radial migration (*Jossin and Cooper, 2011*; *Shah et al., 2017*). In the dentate gyrus, Plexin-A2 negatively regulates Rap1 in migrating neurons (*Zhao et al., 2018*). Rap1 expression has been detected in postnatal CGNs (*Obara et al., 2007*) and therefore it will be interesting in future studies to assess Rap1A/B function in CGN development.

In conclusion, we show here that the timing of expression of Plexin-B2 in CGN precursors in the EGL, sets the pace for cell division and migration and that its downregulation is required for segregating post-mitotic CGNs from cycling precursors. The abnormal association of proliferation and migration in the *Plxnb2* mutant together with an excessive motility probably explain the alteration of foliation and layering observed in *Plxnb2* knockout cerebellum (*Legué et al., 2015*). Interestingly, the transcription factor Zeb1, is also selectively expressed in CGN precursors in the outer EGL and inhibits the CGN differentiation (*Singh et al., 2016*). Moreover, it inhibits the expression of Rnd1 and Rnd3 GTPases. It will be interesting to determine if *Plxnb2* is a target of Zeb1.

# Materials and methods

## Key resources table

| Reagent type (species) or resource | Designation | Source or reference | Identifiers | Additional information |
|---|---|---|---|---|
| Strain; strain background (*Mus musculus*) | *En1*<sup>Cre</sup> (C57BL/6J) | DOI:10.1101/gad.14.11.1377 | *En1*<sup>tm2(cre)Wrst</sup>/J RRID:IMSR_JAX:007916 | |
| Strain; strain background (*Mus musculus*) | *Wnt1-Cre* (C57BL/6J) | DOI:10.1002/dvdy.20611 | B6.Cg-*E2f1*<sup>tg(Wnt1-cre)2Sor</sup>/J RRID:IMSR_JAX:022501 | |
| Strain; strain background (*Mus musculus*) | *Plxnb2*<sup>-/-</sup> | DOI:10.1523/JNEUROSCI.4710-06.2007 | *Plxnb2*<sup>tm1Matl</sup> RRID:MGI:4881705 | Gift from Roland Friedel |
| Strain; strain background (*Mus musculus*) | *Plxnb2* cKO | DOI:10.1523/JNEUROSCI.5381-06.2007 | | |
| Antibody | Anti-Calbindin D-28k (Rabbit antiserum) | Swant | Cat# CB38, RRID:AB_2721225 | IF(1:1000), |
| Antibody | Anti-Calbindin D-28k (Mouse monoclonal) | Swant | Cat# 300, RRID:AB_10000347 | IF(1:1000) |

*Continued on next page*

*Continued*

| Reagent type (species) or resource | Designation | Source or reference | Identifiers | Additional information |
|---|---|---|---|---|
| Antibody | Anti-Foxp2 (N16) (Goat polyclonal) | Santa Cruz | Cat# sc-21069, RRID:AB_2107124 | IF(1:1000) |
| Antibody | Anti-Glial fibrillary acidic protein (GFAP) (Mouse monoclonal) | Millipore | Cat# MAB360, RRID:AB_11212597 | IF(1:500) |
| Antibody | Anti-Green Fluorescent Protein (GFP) (Rabbit polyclonal) | ThermoFisher Scientific | Cat# A-11122, RRID:AB_221569 | IF(1:2000) |
| Antibody | Anti-Green Fluorescent Protein (GFP) (Chicken polyclonal) | Aves | Cat# GFP-1010, RRID:AB_2307313 | IF(1:2000) |
| Antibody | Anti-Phospho-Histone H3(Ser10) (Rabbit polyclonal) | Cell Signaling | Cat# 9701; RRID:AB_331535 | IF(1:1000) |
| Antibody | Anti-Ki67 (Rabbit polyclonal) | Abcam | Cat# Ab15580; RRID:AB_443209 | IF(1:500) |
| Antibody | Anti-Olig-2 (Rabbit monoclonal) | Millipore | Cat# AB9610; RRID:AB_10141047 | IF(1:500) |
| Antibody | Anti-Pax6 (Rabbit polyclonal) | Millipore | Cat# AB2237; RRID:AB_1587367 | IF(1:1000) |
| Antibody | Anti-Plexin-B2 (Armenian Hamster monoclonal) | Novus | Cat# NBP1- 43310; RRID:AB_10006672 | IF(1:1000) |
| Antibody | Anti-Sema6A (Goat polyclonal) | R and D systems | Cat# AF1615; RRID:AB_2185995 | IF(1:500) |
| Antibody | Anti-Contactin-2/TAG1 (Goat polyclonal) | R and D systems | Cat# AF4439; RRID:AB_2044647 | IF (1:500) |
| Antibody | Anti- beta-Tubulin III (Rabbit polyclonal) | Sigma-Aldrich | Cat# T2200; RRID:AB_262133 | IF (1:1000) |
| Antibody | Anti- VGLUT1 (Guinea pig polyclonal) | Millipore | Cat# AB5905; RRID:AB_2301751 | IF (1:3000) |
| Antibody | Donkey Anti-Rabbit IgG (H + L) Alexa Fluor 488 | Jackson Immunoresearch | Cat# 711-545-152; RRID:AB_2313584 | IF (1:750) |
| Antibody | Donkey Anti-Rabbit IgG (H + L) Cy3 | Jackson Immunoresearch | Cat# 711-165-152; RRID:AB_2307443 | IF (1:750) |
| Antibody | Donkey Anti-Rabbit IgG (H + L) Alexa Fluor 647 | Jackson Immunoresearch | Cat# 711-605-152; RRID:AB_2492288 | IF (1:750) |
| Antibody | Bovine Anti-Goat IgG (H + L) Alexa Fluor 647 | Jackson Immunoresearch | Cat# 805-605-180; RRID:AB_2340885 | IF (1:750) |
| Antibody | Donkey Anti-Goat IgG (H + L) Cy3 | Jackson Immunoresearch | Cat# 705-165-147; RRID:AB_2307351 | IF (1:750) |
| Antibody | Donkey Anti-Mouse IgG (H + L) Alexa Fluor 647 | Jackson Immunoresearch | Cat# 715-605-150; RRID:AB_2340862 | IF (1:750) |
| Antibody | Donkey Anti-Chicken IgG (H + L) Alexa Fluor 488 | Jackson Immunoresearch | Cat# 703-545-155; RRID:AB_2340375 | IF (1:750) |

*Continued*

| Reagent type (species) or resource | Designation | Source or reference | Identifiers | Additional information |
|---|---|---|---|---|
| Antibody | Donkey Anti-Chicken IgY (H + L) Cy3 | Jackson Immunoresearch | Cat# 703-165-155; RRID:AB_2340363 | IF (1:750) |
| Antibody | Goat Anti-Armenian Hamster IgG (H + L) Alexa Fluor 488 | Jackson Immunoresearch | Cat# 127-545-160; RRID:AB_2338997 | IF (1:750) |
| Antibody | Goat Anti-Armenian Hamster IgG (H + L) Cy3 | Jackson Immunoresearch | Cat# 127-165-160; RRID:AB_2338989 | IF (1:750) |
| Antibody | Donkey Anti-Guinea Pig IgG (H + L) FITC | Jackson Immunoresearch | Cat# 706-095-148; RRID:AB_2340453 | IF (1:750) |
| Antibody | Donkey Anti-Guinea Pig IgG (H + L) Cy3 | Jackson Immunoresearch | Cat# 706-165-148; RRID:AB_2340460 | IF (1:750) |
| Antibody | Donkey Anti-Goat IgG (H + L) Alexa Fluor 488 | Thermo Fisher Scientific | Cat# A-11055; RRID:AB_25341020 | IF (1:750) |
| Antibody | Donkey Anti-Goat IgG (H + L) Alexa Fluor 555 | Thermo Fisher Scientific | Cat# A21432; RRID:AB_2535853 | IF (1:750) |
| Antibody | Donkey Anti-Mouse IgG (H + L) Alexa Fluor 488 | Thermo Fisher Scientific | Cat# A21202; RRID:AB_141607 | IF (1:750) |
| Other | Hoechst 33342 | Thermo Fisher Scientific | Cat# H3570 | IF (1:1000) |
| Commercial assay or kit | Click-iT EdU Cell Proliferation Kit for Imaging, Alexa Fluor 647 dye | Thermo Fisher Scientific | Cat# C10340 | |
| Chemical compound, drug | Gelatin | VWR Chemicals | Cat# 24350.262 CAS Number: 9000-70-8 | |
| Chemical compound, drug | Thimerosal | Sigma-Aldrich | Cat# T8784-5g CAS Number: 54-64-8 | |
| Chemical compound, drug | TritonX-100 | Sigma-Aldrich | Cat# X100-500ml CAS Number: 9002-93-1 | |
| Chemical compound, drug | SYBR Gold nucleic acid stain | Thermo Fisher Scientific | Thermo Fisher Scientific:S11494 | |
| Software, algorithm | Fiji | NIH | RRID:SCR_002285 | Analysis |
| Software, algorithm | GraphPad Prism | GraphPad | RRID:SCR_002798 | Analysis |
| Software, algorithm | Imaris | Oxford Instruments | RRID:SCR_007370 | Analysis |
| Software, algorithm | iMovie | Apple | http://www.apple.com/fr/imovie/ | Analysis |

## Mouse lines

Full *Plxnb2* knockout mice were obtained by breeding *Plxnb2*[tm1Matl] mouse (RRID:MGI:4881705), harboring a targeted secretory trap mutation between exon 16 and 17 in a CD1 background (*Friedel et al., 2007*). Conditional *Plxnb2* knockout mice (*Deng et al., 2007*), harboring loxP sites encompassing *Plxnb2* exons 19–23, were kept in a C57BL/6 genetic background, as were the *En1*[Cre] (JAX #007916, *En1*[tm2(cre)Wrst]/J) (*Kimmel et al., 2000*) and Wnt1-Cre (JAX #022501, B6.Cg-*E2f1*[tg] [(Wnt1-cre)2Sor]/J *Danielian et al., 1998*) mouse lines. For all experiments with conditional lines, *Plxnb2*[fl/]

$^{fl}$ mice were crossed with heterozygous-*Cre*/homozygous-*floxed* animals to obtain *Cre*-positive mutants and *Cre*-negative control animals in the same litter. Genotypes were determined by PCR from genomic DNA isolated from tail samples. All animal housing, handling and experimental procedures were carried out in accordance to institutional guidelines, approved by the Sorbonne University ethic committee (Charles Darwin). Noon on the day of a vaginal plug was considered E0.5 and date of birth as P0.

## Rotarod

The accelerating Rotarod (Columbus) consists of a horizontal rod, 3 cm in diameter, turning on its longitudinal axis. During the training phase, mice walked on the rod at a rotational speed varying from 4 to 40 rpm for one minute. The mice were then subjected to four trials in which the speed of rotation increased gradually from 4 rpm to 40 rpm over 5 min. Time spent on the rod was recorded and averaged for the four trials. The test was repeated three days in a row with the same procedure on the same animals, except that the training session was performed only on the first day. Animals of both sexes were used since no sex-dependent effect on locomotion was expected.

## In situ hybridization

Sense and antisense RNA probes were designed to cover the floxed exons (19-23) of *Plxnb2* and in vitro transcribed from cDNA encoding the full-length mouse *Plxnb2* gene using the following primers (both including the T7 and T3 RNA polymerase binding sequences respectively: Forward 5'-TAATACGACTCACTATAGGGGCCTTTGAGCCATTGAGAAG −3' and Reverse 5'- AATTAACCCTCACTAAAGGGACGTCATTCGTCTGGTCCTC −3'). Probes were labeled with digoxigenin-11UTP (Roche Diagnostics) and in situ hybridizations were performed as previously described (*Marillat et al., 2004*). Slides were scanned with a Nanozoomer (Hamamatsu), images were handled in NDP.view2 (Hamamatsu) and corrected for brightness and contrast in Photoshop 21.0.2 (Adobe).

## Immunohistochemistry

Pups were deeply anesthetized with ketamine (200 mg/kg) and xylazine (20 mg/kg) before transcardiac perfusion with 4% PFA in PBS. Brains were post-fixed for maximum 24 hr, cryoprotected in 30% sucrose in PBS, embedded in 7.5% gelatin/10% sucrose in 0.12 M phosphate buffer, and frozen in isopentane at −50°C. Cryosections of 20 µm or microtome sections of 50 µm were blocked with PBS-GT (0.2% gelatin, 0.25% Triton-X100 (sigma) in PBS) and incubated overnight at RT with the following primary antibodies against Plexin-B2, Pax6, Calbindin (CaBP), Ki67, Tag1, H3P, GFP, Myelin (MOG), Vglut1 as listed in the Key Resources Table. Species-specific Alexa-conjugated secondary antibodies (Jackson ImmunoResearch or Invitrogen) were diluted 1:750 and incubated 1–2 hr at RT. Sections were counterstained with DAPI and embedded in Mowiol. Images were acquired with a DM6000 epifluorescence microscope (Leica) and a laser scanning confocal microscope (FV1000, Olympus) using Fluoview FV10-ASW software (Olympus). Images were reconstructed using FIJI (NIH) or Imaris software (bitplane). Adobe Photoshop was used to adjust brightness, contrast, and levels.

## Whole mount immunohistochemistry and 3D-light sheet microscopy

Mice were deeply anesthetized with ketamine (200 mg/kg) and xylazine (20 mg/kg) and transcardially perfused with 4% paraformaldehyde in PBS. Brains were briefly post-fixed, dehydrated in serial Methanol dilutions, bleached ON in Methanol + 5% $H_2O_2$ at 4°C and rehydrated. Brains were permeabilized in PBS with 0.2% Gelatin, 0.5% Triton-X100 and 1 mg/ml Saponin before incubation with primary antibodies against Pax6, FoxP2, and/or GFP in the same buffer (see Key Resources Table for antibody RRIDs and dilutions). Secondary antibodies conjugated to Alexa-fluorophores (Jackson ImmunoResearch) were incubated (1:750) together with TO-PRO-3 (1:150, Invitrogen). Tissue clearing was conducted using a methanol dehydration series, dichloromethane-mediated delipidation, and dibenzyl ether immersion, according to the iDISCO+ clearing protocol (*Belle et al., 2014*; *Renier et al., 2014*). 3D-imaging was performed with a fluorescence light sheet ultramicroscope (Ultramicroscope I, LaVision BioTec). 3D volumes were generated using Imaris x64 software (Bitplane). Videos of tiff image-sequences were converted in FIJI and edited in iMovie (Apple).

## EdU labeling and quantification of proliferating cells

Two or 24 hr prior to fixation, pups were injected intraperitoneally with 0.1 ml/10 g bodyweight EdU solution of 5 mg/ml (Invitrogen). Pups were deeply anesthetized with ketamine (200 mg/kg) and xylazine (20 mg/kg) before transcardiac perfusion with 4% PFA in PBS. Brains were post-fixed for 24 hr and cut in 20-µm-thick sections with a cryostat (Leica). Slides were incubated with primary antibody against H3P (see Key Resources Table) followed by Alexa-conjugated secondary antibodies. EdU was revealed using the EdU Click-it imaging kit (Invitrogen). High-resolution images of 3 mid-vermis sections per animal were acquired using an inverted Olympus FV1000 confocal laser-scanning microscope. Mosaic images were stitched using Imaris stitcher (Bitplane). Imaris software (Bitplane) was used to segment straight-cut stretches of lobe V/VI EGL to be used for semi-manual 'spots' cell count of EdU and H3P-positive nuclei in 3D. DAPI signal was used to determine total EGL volume of analyzed area. Mann-Whitney U (MWU) non-parametric statistical analysis was performed using Prism seven software (Graphpad).

## In vivo cerebellum electroporation

P7 or P11 mouse pups were ice-anesthetized and approximately 5 µl of a 1 µg/µl DNA solution (pCX-GFP or pCX-tdTomato) with 0.01% Fast Green was injected subdurally at the level of the cerebellum using a glass needle (FHC PhymEP). CGN precursors in the external granule layer were electroporated with five 50 ms pulses of 120 V with 950 ms interval using a tripolar electrode (*dal Maschio et al., 2012*). The pup's head was held between a platinum-coated tweezer-electrode (positive pole, Harvard Apparatus) and a third platinum electrode (custom-made, negative pole) covered the cerebellar area. After electroporation pups were quickly revived, and placed back in the litter. Two hours prior to transcardiac perfusion, pups were injected intraperitoneally with 0.1 ml/10 g bodyweight EdU (Invitrogen) solution of 5 mg/ml in 0.6% NaCl. Pups were deeply anesthetized with ketamine (200 mg/kg) and xylazine (20 mg/kg) before transcardiac perfusion with 4% PFA in PBS. Brains were post-fixed for 24 hr. Coronal or sagittal 50 µm thick slices were made using a freezing microtome. For tridimensional visualization of the morphology of electroporated CGNs, adult mice electroporated at P7 were transcardially perfused as described, and their brains were processed for whole-mount immunohistochemistry and tissue clearing as described above.

## EGL explant cultures, time-lapse imaging, and immunocytochemistry

P4-5 mouse pups were decapitated, 5 µl of 1 µg/µl pCX-GFP in 0.01% Fast Green was injected subdurally above the cerebellum and CGNs were electroporated as described above. Explants were made from both non-treated and electroporated cerebella (*Kerjan et al., 2005*). Cerebella were rapidly removed and placed in ice-cold L15 dissection medium (Gibco). 350-µm-thick sagittal slices were made using a tissue chopper (MacIlwain) and 300–400 µm blocks of Fast Green containing EGL were dissected. EGL explants were seeded on 100 µg/ml PLL and 40 µg/µl laminin-coated coverslips, or on similarly treated glass-bottom well plates (MatTec corp.). Explants were incubated in BME supplemented with 5% sucrose, 0.5% BSA, ITS, L-Glutamine and Pen/Strep, at 37°C and 5% $CO_2$.

15 min interval time-lapsed acquisitions were taken after 24 hr (One day in vitro (DIV1)) in culture for at least 18 hr with an inverted Olympus FV1000 confocal laser-scanning microscope. Migration of GFP$^+$ cells was analyzed by semi-manual tracking using FIJI software. Total cell division events were counted using differential interference contrast and GFP signal. Mann-Whitney U non-parametric statistical analysis was performed using Prism seven software (Graphpad).

For immunocytochemistry, culture medium was half replaced with medium containing 20 µM EdU (ThermoFischer, 10 µM end concentration) 2 hr prior to fixation by 1:1 addition of 8% paraformaldehyde and 8% sucrose in PBS. Explants were incubated overnight with primary antibodies against GFP βIII-tubulin, Pax6, H3P, Sema6a, GFAP, or Olig2 (antibody IDs and dilutions are listed in Key Resources Table). Secondary antibodies conjugated to Alexa fluorophores (1:750 Jackson ImmunoResearch or Invitrogen) were incubated for 1 hr. EdU was revealed using the Click-it imaging kit (Invitrogen). Images were taken using an inverted Olympus FV1000 confocal laser-scanning microscope and Fluoview FV10-ASW software (Olympus). For quantification of cell counts and neurite length, images were acquired using a DM6000 epifluorescence microscope (Leica). DAPI- and EdU-positive cells were counted using thresholded images and analyzed with FIJI software using particles tool

and/or Sholl analysis tool. Mann-Whitney U (MWU) non-parametric statistical analysis was performed using Prism seven software (Graphpad).

## Data analysis and quantification

All statistical tests were performed by comparing averaged material from at least four different animals or three culture experiment repeats. All specific $N$s and statistical test result information are provided in the Figure legends. Non-parametric Mann-Whitney U tests or student T-test were performed in Prism seven software. $p<0.05$ was considered as statistically significant. All graphs (unless otherwise specified) show individual data points and average ± SEM. All source data used to render the graphs is included in excel files.

Cerebellar volume (*Figure 3—figure supplement 1A*) was measured by manual segmentation in Imaris of 5 to 6 P4 and P30 cerebella from both *Plxnb2$^{fl/fl}$* control and *En1$^{Cre}$;Plxnb2$^{fl/fl}$* mutant animals (paraflocculus was not taken into account since it was sometimes removed during the dissection).

Cerebellar layer thickness (*Figure 4A*) was measured from coronal sections from 11 control and 13 mutant P9 brains. Thickness was measured in FIJI from 1 to 3 sections per brain, and averaged from 12 points per section in mid-cerebellar regions.

The amount of EdU$^+$ and/or H3P or Ki67 nuclei (*Figure 4C,D*, *Figure 4—figure supplement 1A*) was counted semi-automatically using the 'spots' function in Imaris from five control and five mutant animals for each condition. Three sagittal sections (50 µm thick) per brain were imaged at high-magnification with a confocal microscope. The DAPI signal was segmented and used as value for EGL volume in which EdU cells were counted.

GFP-positive CGN cell body measurements, process length, and GFP/EdU or GFP/Pax6 co-staining were assessed with FIJI from 50 µm thick coronal sections imaged at high-magnification with a confocal. Specific N for each experiment can be found in figure legends. Cells with clear leading process(es) were considered uni- or bipolar CGN. Cells with a multipolar or round appearance were considered as CGN precursors. (*Figure 5* and *Figure 5—figure supplement 1*).

DAPI/Calbindin signal in confocal z-stacks of 50 µm thick coronal sections (usually 2–3 sections per brain) was used to equally divide the EGL in two bins since in *Plxnb2* mutant, the EGL did not display clear inner/outer EGL boundary. All GFP$^+$ CGNs were counted and assigned to multipolar or bipolar morphology. Percentages of multipolar cells or bipolar cells were calculated per bin (*Figure 5—figure supplement 1C*).

The relative distribution of parallel fibers in the molecular layer (*Figure 6A*) was measured in FIJI from 50-µm-thick coronal sections and averaged from three different points per cerebellum from 7 *Plxnb2$^{fl/fl}$* and 4 *En1$^{Cre}$;Plxnb2$^{fl/fl}$* cerebella. The thickness of the part of the ML containing GFP$^+$ or tdTom$^+$ parallel fibers was measured and normalized to the total thickness of the ML.

The distribution of Vglut1$^+$ puncta (*Figure 6—figure supplement 1A*) was assessed by measuring the integrated density of the Vglut1 fluorescent signal in 5 distal and 5 proximal $10 \times 10$ µm squares along different Purkinje cells per animal, from high-magnification confocal z-stacks from 20 µm thick sagittal sections. Data was averaged from four animals for both genotypes. A ratio closer to 0 means that the Vglut1 density in the distal square (dendrite tips) is much lower than the Vglut1 density in the proximal square (dendrite trunk). A ratio closer to one means that the Vglut1 signal in the distal square is almost the same as the signal in the proximal region.

The distribution of DAPI$^+$ nuclei in DIV1 EGL explants (*Figure 7B*) was measured by Sholl analysis in FIJI. Cells were counted by the analyze particles tool of FIJI. Multiple t-tests were used to calculate statistical difference between the distance (continuous Sholl circles, 1 µm apart) neurons migrated outward control or *Plxnb2* mutant explants.

The colocalization of DAPI/Pax6, GFP/Pax6, EdU/H3P or GFP/EdU (*Figure 7*, *Figure 7—figure supplements 1* and *2*, *Figure 9*) in DIV1 and/or DIV2 EGL explants was assessed in FIJI from thresholded high-magnification confocal images. GFP$^+$ CGNs were considered bipolar when two clear processes pointing in opposite direction were present. Leading and trailing process length was measured in FIJI. Specific Ns for each experiment can be found in the figure legends.

Time-lapse imaging was started around DIV1 and continued for at least 18 hr (max 27 hr) with a 15-min interval. The tiff image sequence was opened in FIJI and bipolar GFP$^+$ CGNs (*Figure 8*) were traced using semi-manual tracking from the moment they had a clear bipolar morphology until the end of the imaging period or until the moment they left the field of view. Multipolar GFP$^+$ CGNs

(*Figure 10*) were traced from the beginning of the imaging period, or the moment they left the explant, until they divided. These measurements were used to calculate and compare movement and total distance. Time between cytokinesis and the acquisition of a bipolar morphology was also measured in FIJI.

## Acknowledgements

We thank Dr Roland Friedel for providing the *Plxnb2* knockout and Dr Okabe for the pCX plasmid. The authors wish to thank Morgane Belle for invaluable assistance with confocal- and ultramicroscopes and Imaris software, and all members of Alain Chédotal and Xavier Nicol teams for helpful discussions and input during the project. This work was completed with the financial support of the Programme Investissements d'Avenir IHU FOReSIGHT (ANR-18-IAHU-01), and a grant from the Fondation pour la recherche médicale (FRM).

## Additional information

### Competing interests

Rohini Kuner: Reviewing editor, *eLife*. The other authors declare that no competing interests exist.

### Funding

| Funder | Grant reference number | Author |
| --- | --- | --- |
| Agence Nationale de la Recherche | ANR-18-IAHU-01 | Alain Chédotal |
| Fondation pour la Recherche Médicale | CS 2018 | Alain Chédotal |

The funders had no role in study design, data collection and interpretation, or the decision to submit the work for publication.

### Author contributions

Eljo Van Battum, Data curation, Formal analysis, Investigation, Visualization, Methodology, Writing - original draft, Writing - review and editing; Celine Heitz-Marchaland, Data curation, Formal analysis, Investigation, Visualization, Writing - review and editing; Yvrick Zagar, Data curation, Formal analysis, Methodology, Writing - review and editing; Stéphane Fouquet, Data curation, Visualization, Methodology, Writing - review and editing; Rohini Kuner, Resources, Writing - review and editing; Alain Chédotal, Conceptualization, Supervision, Funding acquisition, Methodology, Writing - original draft, Project administration, Writing - review and editing

### Author ORCIDs

Eljo Van Battum https://orcid.org/0000-0002-3358-5039
Alain Chédotal https://orcid.org/0000-0001-7577-3794

### Ethics

Animal experimentation: All animal housing, handling and experimental procedures were carried out in accordance to institutional guidelines, approved by the Sorbonne University ethic committee (Charles Darwin, permit 03787.02).

### Decision letter and Author response

Decision letter https://doi.org/10.7554/eLife.60554.sa1
Author response https://doi.org/10.7554/eLife.60554.sa2

## Additional files

### Supplementary files
• Transparent reporting form

### Data availability
All data generated or analysed during this study are included in the manuscript and supporting files.

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
