## [Decision Letter]

**Acceptance summary:**

Your study provides new information on Plexin-B2, an unusual Plexin in the CNS as its expression is restricted to neuronal precursors in the subventricular zone and external granule layer (EGL) of the cerebellum. Your previous study showed that cerebellum foliation and layering are severely affected in full Plexin-B2 knockout. Now with a cerebellum-specific knockout, you have used use electroporation to label and follow cells in the EGL and elsewhere, 3D imaging, tissue clearing and time-lapse video-microscopy of cerebellum explants. With superb and precise views of the wild type and mutant phenotype, you demonstrate effects of perturbing Plexin-B2 on tangential progenitor cell migration, proliferation, radial migration, and parallel fiber organization.

In this revision, viewed favorably by all three reviewers, you clarified that the scattered parallel fibers in the Plxnb2 mutant were likely made due to the abnormal motility of early CGNs not only within the EGL, but also into the molecular layer (ML). Your new analysis of mitosis in bipolar GFP cells also supports the existence (but not prominence) of mitotically active and tangentially migrating CGNs in the mutant. And, while the cell cycle has not been so readily assessed in vivo, you have nicely depicted the migration of cells and the positioning of parallel fibers as a consequence of the misplacement of the granule cells. This study illuminates new criteria for cerebellar development, still being uncovered with approaches such as those utilized for your study.

**Decision letter after peer review:**

Thank you for submitting your article "Plexin-B2 controls cell cycle exit and motility of cerebellar granule neurons" for consideration by *eLife*. Your article has been reviewed by 3 peer reviewers, and the evaluation has been overseen by a Reviewing Editor and Marianne Bronner as the Senior Editor. The following individuals involved in review of your submission have agreed to reveal their identity: Roman J Giger (Reviewer #1); Keiko Yamamoto (Reviewer #2); David J Solecki (Reviewer #3).

The reviewers have discussed the reviews with one another and the Reviewing Editor has drafted this decision to help you prepare a revised submission.

Summary:

Your paper showing that one mammalian Plexin, Plexin-B2, is unusual in the CNS, as its expression is restricted to neuronal precursors in the subventricular zone and egl of the cerebellum. Your previous study showed that cerebellum foliation and layering are severely affected in full Plexin-B2 knockout. Now with two cerebellum-specific ko's, you have offset the poor viability in these mice, and use electroporation to label and follow cels in the egl, by 3D imaging, tissue clearing and time-lapse video-microscopy of cerebellum explants. These approaches have produced excellent and precise views of the mutant phenotype, from which you cite effects of reducing Plexin-B2 on cell division and the maintenance of undifferentiated granule cells progenitors in their niche. You indicate that granule cell precursors are more motile, their migration is perturbed, and their differentiation delayed, resulting in a severe disorganization of granule cell positioning but and parallel fiber layering.

All three reviewers thought that your paper was of great interest, well executed in the main, and that it provides new information on Plexin-B2, but that further analyses of your data would better explain your claims. We feel that your findings related to granule neuron progenitor cell cycle length as it pertains to the timing of maturation, and related to this, the effects on migration, and on parallel fiber extension and positioning, need additional characterization. These experiments should be doable, provided you have the mice, and in through additional analyses of your existing images. Details of necessary revisions are included below.

Essential revisions:

1. To verify that granule cell precursors are more motile in the mutants, yet still presumably capable of proliferation, but having delayed differentiation, the following amendments are suggested:

a. The current conclusion of extended cell cycle length is the least developed of your new findings. One could interpret Tag1 positive cells that are also Ki67 positive as precocious differentiation, instead of the interpretation provided. Standard assays like assessing the balance of actively proliferating progenitors via Ki67 and EdU double labeling and an assessment of cell cycle fractions via FACS to determine DNA conent of Dapi or Hoechst labeled would address this concern.

b. In Figure 5D, you show that EdU+ and GFP+ tangentially migrating bipolar CGNs are increased in Plxnb2 mutants, and interpreted these results to suggest that a significant fraction of tangentially migrating CGNs are mitotically active in the EGL of Plxnb2 mutants (lines 476-477). Are they are truly mitotic? Because EdU was administered 2 hours before the fixation, the differentiation step to tangentially migrate GCNs after the last mitosis might be more rapid in Plxnb2 mutants than in control EGL. This could be clarified by PH3 immunohistochemistry, that would indicate mitotic cells at the time of fixation, for comparison.

c. It would be good to see Tag1+ co co-stained cells that have the EdU and GFP signal to confirm that migrating cells are also proliferative.

2. To further demonstrate effects on motility of early CGNs in the mutants: You show that cells labeled with Ki67 and Tag1 are intermingled (Figure 4B), and conclude that motility of CGN precursors and tangentially migrating CGNs is altered in Plexin-B2 mutants. However, it is not clear how these aspects are altered. For example, even though the motility is altered in Plexin-B2 mutants, nicely shown in the explants, are tangentially migrating CGNs and CGN precursors in vivo still located within EGL, or are some of them located in the ML? Are both tangentially migrating CGNs and CGN precursors, or only CGN precursors ectopically localized? With your excellent technique of in vivo electroporation, and visualizing individual cells with GFP, in addition to quantitative analysis of morphological features shown in Figure 5C-E, localization of GFP-positive tangentially migrating CGNs and CGN precursors in the oEGL, iEGL, or ML could be quantitatively analyzed. This would a better connection to the scattered parallel fibers in Plexin-B2 mutants (Figure 6A). This could be done with the images you already have in hand.

3. Further analysis of parallel fiber organization, particularly for the scattered parallel fibers across the ML (Figure 6A): It is difficult to see the connection with the phenotypes of the earlier developmental steps and to picture how this happens. One possibility would be that Plxnb2 mutant parallel fibers move in either direction, close to and far from Purkinje cell somas, during or after CGNs radially migrate and parallel fibers extend. Another possibility would be that the scattered parallel fibers are due to dispersion of tangentially migrating CGN to the oEGL and to the ML. You could address these questions by:

a. Quantitative analysis of distribution of tangentially migrating CGNs in the ML, oEGL, and iEGL.

b. Quantitative analysis or clearer images of GFP-positive parallel fibers in the ML during development: Although you mention that in P9 Plxnb2 mutants, the young GFP+ CGN 'axonal' fibers fun throughout the entire ML (lines 306-307), it is not clear whether GFP+ signals in Figure 6B are parallel fibers or ascending axons.

c. By in vivo electroporation at P11 and to observe at P13, at which time the ML is wider than at P9. This (c.) is an optional suggestion.

4. The experiments using cultured EGL are powerful for directly observing the movement of CGNs. Even though authors referred to studies showing that CGNs in culture follow developmental process resembling in-vivo CGNs (lines 318-320), additional explanation regarding how they resemble in-vivo CGNs would help to understand the results obtained from cultured. For example, do "multipolar" and "bipolar" CGNs in culture correspond to mitotic and tangentially migrating CGNs in vivo, respectively. If so, can you relate this correspondence either from the literature or by your own analyses authors, in the Methods and Results text.

---

## [Author Response]

Essential revisions:1. To verify that granule cell precursors are more motile in the mutants, yet still presumably capable of proliferation, but having delayed differentiation, the following amendments are suggested:a. The current conclusion of extended cell cycle length is the least developed of your new findings. One could interpret Tag1 positive cells that are also Ki67 positive as precocious differentiation, instead of the interpretation provided. Standard assays like assessing the balance of actively proliferating progenitors via Ki67 and EdU double labeling and an assessment of cell cycle fractions via FACS to determine DNA conent of Dapi or Hoechst labeled would address this concern.

After looking carefully our text we think that we were unclear and that there was some misunderstanding: we wrote “In control P9 cerebellum, both markers were segregated (Figure 4B), whereas in Plxnb2 mutants, CGN precursors lost their confinement to the oEGL and intermingled with tangentially migrating Tag1^+^ CGNs in the iEGL (Figure 4B)”

We did not intend to say here that some Ki67+ were Tag1 + cells and now make this clear in the text: “However, as Ki67 and Tag1 are expressed in different cell compartments (nucleus and cell surface respectively), we could not determine if some of the Tag1+ cells were also Ki67+”.

However, as requested in comment 1c we have performed an experiment to show the co-expression of a mitotic marker (H3P) and Tag1 in tangential bipolar GFP+ CGNs. This suggests that such double positive cells exist. (Figure 5F).

We also fully agree with the reviewer that it is still difficult to conclude on a possible alteration of the cell cycle.

We could not perform the FACS analysis as we don’t have enough mice (colonies had to be downsized due to the 2 successive and ongoing lockdowns). Moreover, we don’t really know how it could be done in this neuronal system as we could not find papers using this type of procedure to study cell cycle in dissociated granule cell progenitors.

However, we have performed the requested double labeling for Ki67 and EdU on P8 cerebellum sections instead, to estimate differences in cell cycle rate. The quantification of the proportion of Edu+/Ki67+ cells did not show any statistically-significant difference between controls and *En1cre;Plxnb2fl/fl* mutants (Figure 4—figure supplement 1B) confirming what we had seen before with Ki67/BrDU double labelling in complete Plxnb2-/- KO (Friedel et al., J Neurosci. 2007). Therefore, it is indeed difficult at this stage to conclude about cell-cycle progression and we now say:

“Together these results suggest that there is probably no alteration of cell cycle progression in absence of Plexin-B2 although more experiments will be required to determine if the M phase is affected”.

We have also modified the title of the paper accordingly.

b. In Figure 5D, you show that EdU+ and GFP+ tangentially migrating bipolar CGNs are increased in Plxnb2 mutants, and interpreted these results to suggest that a significant fraction of tangentially migrating CGNs are mitotically active in the EGL of Plxnb2 mutants (lines 476-477). Are they are truly mitotic? Because EdU was administered 2 hours before the fixation, the differentiation step to tangentially migrate GCNs after the last mitosis might be more rapid in Plxnb2 mutants than in control EGL. This could be clarified by PH3 immunohistochemistry, that would indicate mitotic cells at the time of fixation, for comparison.

We agree with the reviewer that a more rapid differentiation could also explain the presence of GFP+/EdU+ cells.

We performed H3P and EdU staining on P8 sections of brains that were electroporated at P7 and injected with EdU 2h before isolation in both control and *En1cre;Plxnb2fl/fl* animals. Although H3P/EdU double-positive bipolar GFP cells were very rare in the mutant, we never observed any of them in the control brains. We show a few compelling examples in Figure 5—figure supplement 1D.

However, as mentioned in point 1a, we also present the alternative explanation (faster differentiation) in the revised manuscript.

c. It would be good to see Tag1+ co co-stained cells that have the EdU and GFP signal to confirm that migrating cells are also proliferative.

We have performed this experiment and included it in Figure 5F (replacing 5C). Instead of 2h EDU we use the mitotic marker H3P. The panels in Figure 5F show co-labeling of GFP and H3P in proliferating CGNs. Tag1 marks tangential cells and we can find H3P in these bipolar cells (GFP/Tag1+) only in *Plxnb2* mutant brains.

2. To further demonstrate effects on motility of early CGNs in the mutants: You show that cells labeled with Ki67 and Tag1 are intermingled (Figure 4B), and conclude that motility of CGN precursors and tangentially migrating CGNs is altered in Plexin-B2 mutants. However, it is not clear how these aspects are altered. For example, even though the motility is altered in Plexin-B2 mutants, nicely shown in the explants, are tangentially migrating CGNs and CGN precursors in vivo still located within EGL, or are some of them located in the ML? Are both tangentially migrating CGNs and CGN precursors, or only CGN precursors ectopically localized? With your excellent technique of in vivo electroporation, and visualizing individual cells with GFP, in addition to quantitative analysis of morphological features shown in Figure 5C-E, localization of GFP-positive tangentially migrating CGNs and CGN precursors in the oEGL, iEGL, or ML could be quantitatively analyzed. This would a better connection to the scattered parallel fibers in Plexin-B2 mutants (Figure 6A). This could be done with the images you already have in hand.

To assess the position of multipolar and bipolar cells in the EGL and ML we have performed an additional analysis. The EGL was divided in two equal bins (bin 1 at the surface, encompassing roughly the oEGL, bin 2 at the lower part). Cells residing in the ML were counted as such. Because cells were only slightly dipping into the ML at this time, no additional bins were made within the ML. In controls, GFP+ cells were never found anywhere lower than at the top of the ML. As expected, multipolar progenitors reside in the upper part of the EGL, and bipolar cells are mostly migrating in the lower part of the EGL, and sometimes have just entered the ML. In the case of *En1cre;Plxnb2fl/fl* mutant mice, multipolar and bipolar cells are distributed more evenly over both the upper and lower bins of the EGL. Although many bipolar cells dip into the ML, multipolar GFP+ progenitors were never found there. The graph and example panels of the EGL bins can be found in Figure 5—figure supplement 1C.

3. Further analysis of parallel fiber organization, particularly for the scattered parallel fibers across the ML (Figure 6A): It is difficult to see the connection with the phenotypes of the earlier developmental steps and to picture how this happens. One possibility would be that Plxnb2 mutant parallel fibers move in either direction, close to and far from Purkinje cell somas, during or after CGNs radially migrate and parallel fibers extend. Another possibility would be that the scattered parallel fibers are due to dispersion of tangentially migrating CGN to the oEGL and to the ML. You could address these questions by:a. Quantitative analysis of distribution of tangentially migrating CGNs in the ML, oEGL, and iEGL.

We think this is similar to Comment 2 above and have added this quantification which confirm the “invasion” of the inner EGL and ML by tangentially migrating CGNs in Plxnb2 mutants.

b. Quantitative analysis or clearer images of GFP-positive parallel fibers in the ML during development: Although you mention that in P9 Plxnb2 mutants, the young GFP+ CGN 'axonal' fibers fun throughout the entire ML (lines 306-307), it is not clear whether GFP+ signals in Figure 6B are parallel fibers or ascending axons.

As requested by this reviewer we are now providing on Figure 6B additional images which better illustrate this phenotype. We have chosen to present coronal sections as parallel fibers elongate along the medio-lateral axis (the initial sagittal panels have been moved to Figure 6-figure supplement 1). Therefore, parallel fiber morphology and location are easier to assess. In controls, the characteristic T-shape morphology normally emerged at the border between the iEGL and ML, close to the upper tips of the developing Purkinje cell dendrites. As development progresses, the dendritic tips grow past this point, and parallel fibers grow further in both directions of the ‘T’. By contrast, in Plxnb2 mutant, the T-shape, despite being as straight as in controls, is sometimes formed in the middle of the EGL or even lower in the ML, and the fibers extend in random positions within the ML, EGL or even past Purkinje cell bodies. Furthermore, Therefore, young (maximum 48h-old) parallel fibers enter the molecular layer and overlap with Purkinje dendrites in the *Plxnb2* mutant, whereas in controls, parallel fibers start extending in the iEGL above the tip of the Purkinje cell dendrites.

c. By in vivo electroporation at P11 and to observe at P13, at which time the ML is wider than at P9. This (c.) is an optional suggestion.

We performed the requested experiment. Mice were electroporated at P11 and their cerebellum was collected at P13 to see how the growing parallel fibers occupy the developing ML in the *En1cre;Plxnb2* mutant. Indeed, normally the new fibers lie just above the distal tips Purkinje dendrites, at P13. However, in the mutant we see that fibers run across the Purkinje dendrites. The tips of Purkinje dendrites sometimes reach far in the EGL. These new data are presented in Figure 6C.

4. The experiments using cultured EGL are powerful for directly observing the movement of CGNs. Even though authors referred to studies showing that CGNs in culture follow developmental process resembling in-vivo CGNs (lines 318-320), additional explanation regarding how they resemble in-vivo CGNs would help to understand the results obtained from cultured. For example, do "multipolar" and "bipolar" CGNs in culture correspond to mitotic and tangentially migrating CGNs in vivo, respectively. If so, can you relate this correspondence either from the literature or by your own analyses authors, in the Methods and Results text.

Although we had already cited a few papers (Kawaji et al., 2004; Kerjan et al., 2005) describing the EGL explant model and how it recapitulates the different steps of granule cell development, we agree that we did not make clear enough in the text that bipolar cells correspond to tangentially migrating granule cells and that multipolar cells to dividing granule cell precursors. This is supported by our data and earlier work such as Yacubova and Komuro, 2002 (now cited in the revised text). This has been corrected in the revised text.